# Learning Mixtures of Unknown Causal Interventions

**Abhinav Kumar**
LIDS, Massachusetts Institute of Technology
Broad Institute of MIT and Harvard
`akumar03@mit.edu`

**Kirankumar Shiragur**
Microsoft Research
`kshiragur@microsoft.com`

**Caroline Uhler**
LIDS, Massachusetts Institute of Technology
Broad Institute of MIT and Harvard

## Abstract

The ability to conduct interventions plays a pivotal role in learning causal relationships among variables, thus facilitating applications across diverse scientific disciplines such as genomics, economics, and machine learning. However, in many instances within these applications, the process of generating interventional data is subject to noise: rather than data being sampled directly from the intended interventional distribution, interventions often yield data sampled from a blend of both intended and unintended interventional distributions.

We consider the fundamental challenge of disentangling mixed interventional and observational data within linear Structural Equation Models (SEMs) with Gaussian additive noise without the knowledge of the true causal graph. We demonstrate that conducting interventions, whether do or soft, yields distributions with sufficient diversity and properties conducive to efficiently recovering each component within the mixture. Furthermore, we establish that the sample complexity required to disentangle mixed data inversely correlates with the extent of change induced by an intervention in the equations governing the affected variable values. As a result, the causal graph can be identified up to its interventional Markov Equivalence Class, similar to scenarios where no noise influences the generation of interventional data. We further support our theoretical findings by conducting simulations wherein we perform causal discovery from such mixed data.

## 1 Introduction

Interventions are experiments that can help us understand the mechanisms governing complex systems and also modify these systems to achieve desired outcomes [1, 5, 21]. For example, in causal discovery, interventions are used to infer causal relationships between variables of interest, which has applications in various fields such as biology [5, 12], economics [8], and psychology [13, 21].

Given their extensive applications, there has been significant research in developing methods and experimental design strategies to conduct interventions under different scenarios [32, 24]. Despite significant efforts to develop sophisticated experimental techniques to perform interventions, they often encounter noise [4, 6, 30]. For example, the CRISPR technology, extensively used to perform gene perturbations (or interventions), is known to have off-target effects, meaning the interventions do not always occur on the intended genes [6, 30]. Consequently, in many applications, performing interventions generates data from a mixture of intended and unintended interventional distributions. Analyzing such mixed data directly can lead to incorrect conclusions, adversely affecting downstream applications. Hence, it is essential to disentangle the mixture and recover the components corresponding to each individual intervention for further use in downstream tasks like causal discovery.

38th Conference on Neural Information Processing Systems (NeurIPS 2024).

In our work, we formally address the challenge of disentangling mixtures of unknown interventional and observational distributions within the framework of linear structural equation models (Linear-SEM) with additive Gaussian noise. Given iid samples from a mixture with a fixed number of components as input, we present an efficient algorithm that can learn each individual component. Our results are applicable to both do and more general soft interventions.

We chose to study our problem in the Linear-SEM with additive Gaussian noise framework for its fundamental importance in the causal discovery literature. Shimizu et al. [25] showed that observational data is sufficient for learning the underlying causal graph when the data-generating process is a Linear-SEM with additive non-Gaussian noise with no latent confounders. However, in the same setting with Gaussian noise, the causal graph is only identifiable up to its Markov Equivalence Class (MEC) [18, 25]. Thus, performing interventions (possibly noisy) is necessary to identify the causal graph, making it an interesting framework for our problem.

**Our contributions**    First, we show that given samples from a mixture of unknown interventions within the framework of Linear-SEM with additive Gaussian noise, there exists an efficient algorithm to uniquely recover the individual components of the mixture. The sample complexity of our procedure scales polynomially with the dimensionality of the problem and inversely polynomially with the accuracy parameter and the magnitude of changes induced by each intervention. Our findings indicate that the recovery error for each individual interventional distribution approaches zero as the number of samples increases. Therefore, in the infinite sample regime, we can recover the true interventional distributions, even when the targets of the interventions are unknown. Second, if the input distributions satisfy a strong interventional faithfulness assumption (as defined in Squires et al. [26]), we can utilize the results from [26] to identify the targets of the interventions, thereby enabling causal discovery using these accurately recovered interventional distributions. Finally, we conduct a simulation study to validate our theoretical findings. We show that as sample size increases, one can recover the mixture parameters, identify the unknown intervention targets, and learn the underlying causal graph with high accuracy.

## 2    Prior Work

**Mixture of DAGs and Interventions.**    There has been a lot of interest in understanding the mixture distribution arising from a collection of directed acyclic graphs (DAGs) [27, 23, 29, 14, 9, 28]. Saeed et al. [23] studied the distribution arising from a mixture of multiple DAGs with common topological order, thus a generalization of our problem. Their method identifies the variables whose conditional distribution across DAGs varies. However, there is no theoretical guarantee for the identifiability of the mixture's components. Like us, Kumar and Sinha [14] also studied the mixture arising from interventions on a causal graph and gives an algorithm to identify the mixture component. However, they assume knowledge of the correct topological order and sample access to the observational distribution. Thiesson et al. [28] also studied the problem of learning a mixture of DAGs using the Expectation-Maximization (EM) framework. However, there is no theoretical guarantee about the identifiability of individual components.

**Learning Mixture of Gaussians.**    Learning a mixture of Gaussians is a heavily studied problem [2, 7, 15, 11]. There exist efficient algorithms both in terms of runtime and sample complexity for a fixed number of components in the mixture [11, 15, 2]. Ge et al. [7] gave an efficient algorithm for the case when the number of components is almost $\sqrt{n}$ where $n$ is the dimension of the variables. However, this method only guarantees identifiability in the perturbative setting, where the true parameters are randomly perturbed before the samples are generated.

**Causal Discovery with Unknown Interventions.**    Recent years have seen the development of methods that perform causal discovery with observational and multiple unknown interventional data [16, 26, 10]. Squires et al. [26] takes multiple but segregated datasets from unknown interventional distributions and aims to identify the unknown interventions and learn the underlying causal graph up to its interventional MEC (I-MEC). Jaber et al. [10] considers the same problem of causal discovery with unknown soft interventions in non-Markovian systems (i.e., latent common cause models). However, they also assume that the interventional and/or observational data is already segregated. To our knowledge, no prior works consider directly learning the causal graph from a mixture of interventions.

# 3  Notation and Preliminaries

**Notation.**  We use the upper-case letter $X$ to denote a random variable and lower-case "$x$" to denote the value taken by the random variable $X$. Let, the uppercase bold-face letter $\boldsymbol{X}$ denote a set of random variables and the lowercase bold-face letter $\boldsymbol{x}$ denote the corresponding value taken by $\boldsymbol{X}$. Let the conditional probability function $\mathbb{P}(\boldsymbol{X} = \boldsymbol{x}|\boldsymbol{Y} = \boldsymbol{y})$ be denoted by $\mathbb{P}(\boldsymbol{x}|\boldsymbol{y})$. We use the calligraphic letter $\mathcal{S}$ to denote a set and $|\mathcal{S}|$ to denote the cardinality of the set $\mathcal{S}$. Let [n] denote the set of natural numbers $\{1, \ldots, n\}$. For any vector $\boldsymbol{v}$, we use the notation $[\boldsymbol{v}]_j$ to denote it's $j^{th}$ entry and for any matrix $M$ we use $[M]_{i,j}$ to denote the entry in the $i^{th}$ row and $j^{th}$ column of $M$. We use $\mathbb{R}_+$ to denote positive scalars.

**Structural Equation Model (SEM)**  Following Definition 7.1.1 in Pearl [18], let a causal model (or SEM) be defined by a 3-tuple $\mathcal{M} = \langle \boldsymbol{V}, \boldsymbol{U}, \mathcal{F} \rangle$, where $\boldsymbol{V} = \{V_1, \ldots, V_n\}$ denotes the set of observed (endogenous) variables and $\boldsymbol{U} = \{U_1, \ldots, U_n\}$ denotes the set of unobserved (exogenous) variables that represent noise, anomalies or assumptions. Next, $\mathcal{F}$ denotes a set of $n$ functions $\{f_1, \ldots, f_n\}$, each describing the causal relationships between the random variables having the form:

$$v_i = f_i(\boldsymbol{pa}(\boldsymbol{V}_i), \boldsymbol{u}_i),$$

where $\boldsymbol{U}_i \subseteq \boldsymbol{U}$ and $\boldsymbol{pa}(V_i) \subseteq \boldsymbol{V}$ are such that the associated causal graph (defined next) is acyclic. A causal graph $\mathcal{G}_{\mathcal{M}}$ [1] is a directed acyclic graph (DAG), where the nodes are the variables $\boldsymbol{V}$ and the edges $\boldsymbol{U}$ with edges pointing from $\boldsymbol{pa}(V_i)$ to $V_i$ for all $i \in [n]$.

**Linear-SEM (with causal sufficiency)**  In this work, we study a special class of such causal models (Gaussian Linear-SEMs) where the function class of each $f_i$ is restricted to be linear and of the form

$$v_i = f_i(\boldsymbol{pa}(V_i), \boldsymbol{u}_i) = \sum_{v_j \in \boldsymbol{pa}(V_i)} \alpha_{ij} v_j + u_i,$$

where $\alpha_{ij} \neq 0, \forall V_j \in \boldsymbol{pa}(V_i)$. The *causal Sufficiency* assumption states that $\boldsymbol{U}_i = \{U_i\}$, i.e., $U_i$ is the only exogenous variable that causally affects the endogenous variable $V_i$. This is equivalent to the absence of any latent confounder (Chapter 9 in [20]). In our work, we consider causally-sufficient Linear-SEMs; with a slight abuse of nomeclature, we will call them Linear-SEMs. The functional relationship between the exogenous and endogenous variables is deterministic, and the system's stochasticity comes from a probability distribution over the exogenous noise variables $\boldsymbol{U}$. Thus, the probability distribution over the exogenous variable $\mathbb{P}(\boldsymbol{U})$ defines a probability distribution over the endogenous variable $\mathbb{P}(\boldsymbol{V})$. Without loss of generality, let the nodes $\{V_1, \ldots, V_n\}$ of the underlying causal graph be topologically ordered. Then, we can equivalently write the above set of equations as:

$$\boldsymbol{v} = A\boldsymbol{v} + \boldsymbol{u} \implies \boldsymbol{v} = (I - A)^{-1}\boldsymbol{u}, \tag{1}$$

where $A$, with $A_{ij} = \alpha_{ij}$, contains the causal effects between the endogenous variables. Thus, the matrix $A$, hereafter described as the adjacency matrix, characterizes the causal relationships between the endogenous variables ($\boldsymbol{V}$), where $A_{ij} \neq 0$ denotes an edge between the variable $V_i$ and $V_j$ in $\mathcal{G}$.

**Linear-SEM with additive Gaussian Noise**  We further specialize the exogenous variable $u_i$ (henceforth referred to as noise variable) to be Gaussian with mean $\mu_i$ and variance $\sigma_i$, i.e. $u_i \sim \mathcal{N}(\mu_i, \sigma_i)$. Thus, the joint distribution of the exogenous variables is given by a multivariate Gaussian distribution $\boldsymbol{u} \sim \mathcal{N}(\boldsymbol{\mu}, D)$ where $[\boldsymbol{\mu}]_i = \mu_i$ and the covariance is given by a diagonal matrix $D$ with $[D]_{ii} = \sigma_i$. Thus, the endogenous variables also follow a multivariate Gaussian distribution with $\mathbb{P}(\boldsymbol{v}) = \mathcal{N}(\boldsymbol{m}, S)$, where $\boldsymbol{m} \triangleq B\boldsymbol{\mu}_i$ $S \triangleq BDB^T$ and $B \triangleq (I - A)^{-1}$. Causal discovery aims to identify the unknown adjacency matrix $A$ given observational or other auxiliary data.

**Interventions.**  Following Definition 7.1.2 from Pearl [18], the new causal model describing the interventional distribution, where the variables in a set $\boldsymbol{I}$ are set to a particular value, is given by $M_{\boldsymbol{I}} = \langle \boldsymbol{U}, V, \mathcal{F}_{\boldsymbol{I}} \rangle$, where $\mathcal{F}_{\boldsymbol{I}} = \{f_j : V_j \notin \boldsymbol{I}\} \cup \{f_i' : V_i \in \boldsymbol{I}\}$ and the functional relationship of every node $V_i \in \boldsymbol{I}$ with their parents and corresponding exogenous variable $U_i$ is changed from $f_i$ to $f_i'$. In particular, the functional relationship of node $V_i$ is changed to

$$v_i = \sum_{v_j \in \boldsymbol{pa}(V_i)} \alpha_{ij}' v_j + u_i' \tag{2}$$

---

[1] We will drop the subscript $\mathcal{M}$ when it is clear from context

where $u_i{}' \sim \mathcal{N}(\mu_i{}', \sigma_i{}')$. Such interventions are broadly referred to as "soft". Several other kinds of interventions are also defined in the literature, e.g., *do, uncertain, soft* etc. [3]. We consider three different types of widely studied specializations of soft interventions in our work:

(1) *shift*: the mean of the noise distribution is shifted by a particular value, i.e., $\mu_i{}' = \mu_i + \kappa$ for some $\kappa \in \mathbb{R}$, and everything else remains the same, i.e., $\sigma_i' = \sigma_i$ and $a_{ij}{}' = a_{ij}, \forall j \in [n]$.

(2) *stochastic do* (henceforth referred as *stochastic*): where all the incoming edges from parents are broken, i.e., $\alpha_{ij}{}' = 0$, and $u_i' \sim \mathcal{N}(\mu_i{}', \sigma_i{}')$.

(3) *do*: in addition to breaking all incoming edges, i.e., $\alpha_{ij}{}' = 0$, we also set the variance of the noise distribution to 0 and the mean to any value of choice, i.e., $u_i \sim \mathcal{N}(\mu_i{}', 0)$.

**Atomic Interventions**  In this work, we consider soft interventions where only one node is intervened at a time, i.e., $|\boldsymbol{I}| = 1$. Thus after a soft intervention on node $V_i$, the adjacency matrix is modified such that $A_i \triangleq A - \boldsymbol{e}_i(\boldsymbol{a}_i - \boldsymbol{a}_i')^T = A - \boldsymbol{e}_i \boldsymbol{c}_i^T$ [2], where $\boldsymbol{c}_i^T \triangleq (\boldsymbol{a}_i - \boldsymbol{a}_i')^T$, $\boldsymbol{a}_i^T$ is the $i^{th}$ row of matrix A and $\boldsymbol{a}_i'^T$ is the new row after intervention such that $[\boldsymbol{a}_i]_k = 0, \forall k \geq i$, and $\boldsymbol{e}_i$ is the unit vector with entry 1 at the $i^{th}$ position and 0 otherwise. Thus, the linear SEM from Eq. 1 is:

$$\boldsymbol{v}_i = (I - A_i)^{-1}\boldsymbol{u}_i = (I - A + \boldsymbol{e}_i \boldsymbol{c}_i^T)^{-1}\boldsymbol{u}_i, \qquad (3)$$

where $\boldsymbol{u}_i \sim \mathcal{N}(\boldsymbol{\mu}_i, D_i)$, $\boldsymbol{\mu}_i = \boldsymbol{\mu} + \gamma_i \boldsymbol{e}_i$ for some $\gamma_i \in \mathbb{R}$, $D_i = D - \delta_i \boldsymbol{e}_i \boldsymbol{e}_i^T$ where $\delta_i \triangleq (\sigma_i - \sigma_i')$, is a diagonal matrix with the $i^{th}$ diagonal entry as $\sigma_i'$ and rest is same as $D$. Thus, the interventional distribution of the endogenous variables is also a multivariate Gaussian distribution, i.e., $\mathbb{P}_i(\boldsymbol{V}) = \mathcal{N}(\boldsymbol{m}_i, S_i)$ where $\boldsymbol{m}_i \triangleq B_i\boldsymbol{\mu}_i$, $S_i \triangleq B_i D_i B_i^T$ and $B_i \triangleq (I - A + \boldsymbol{e}_i \boldsymbol{c}_i^T)^{-1}$.

## 4  Problem Formulation and Main Results

In §4.1, we begin by formulating the problem of learning the mixture of interventions and then state our main result on the identifiability of parameters of the mixture. As a consequence of our identifiability result, under an interventional faithfulness assumption (Squires et al. [26]), we show in §4.2 that the underlying true causal graph can be identified up to its I-MEC using a mixture of unknown interventions, thereby obtaining the same identifiability results as in the unmixed setting.

### 4.1  Learning a Mixture of Interventions

We begin by formally defining the mixture of interventions over Linear-SEM with additive Gaussian noise and then state the main result of our paper — a mixture of interventions can be uniquely identified under a mild assumption discussed below.

**Definition 4.1** (Mixture of Soft Atomic Interventions).  *Let $\mathcal{M} = \langle \boldsymbol{V}, \boldsymbol{U}, \mathcal{F} \rangle$ be an unknown Gaussian Linear SEM where the distribution of the endogenous variables is given by $\mathbb{P}(\boldsymbol{V})$ (see §3). Let $\mathcal{I} = \{i_1, \ldots, i_K\}$, hereafter referred to as intervention target set, be a set of unknown soft atomic interventions where each $i_k$ generates a new interventional distribution $\mathbb{P}_i(\boldsymbol{V})$. Then, the mixture of soft atomic intervention is defined as:*

$$\mathbb{P}_{mix}(\boldsymbol{V}) = \sum_{i_k \in \mathcal{I}} \pi_{i_k} \mathbb{P}_{i_k}(\boldsymbol{V}) \qquad (4)$$

*where $\pi_{i_k} \in \mathbb{R}_+$, henceforth referred to as mixing weight, is a positive scalar such that $\sum_{i_k \in \mathcal{I}} \pi_{i_k} = 1$, $i_k \in [n] \cup \{0\}$ where $n = |\boldsymbol{V}|$ is the number of endogenous variables. We also allow $i_k = 0$, which denotes the setting when none of the nodes is intervened, i.e., $P_0(\boldsymbol{V}) \triangleq \mathbb{P}(\boldsymbol{V})$. Using Eq. 1 and 3, a mixture defined over a linear SEM with additive Gaussian noise is a mixture of Gaussians with parameters $\theta = \{(\boldsymbol{m}_{i_k}, S_{i_k}, \pi_{i_k})\}_{i_k \in \mathcal{I}}$ where $\mathbb{P}_{i_k}(\boldsymbol{V}) = \mathcal{N}(\boldsymbol{m}_{i_k}, S_{i_k})$.*

Having defined a mixture of interventions, we then aim to answer the following questions: (1) Does there exist an algorithm that can uniquely identify the parameters ($\theta$) of the mixture of interventions under an infinite sample limit? (2) What is the run time and sample complexity of such an algorithm? It is immediate that if the intervention doesn't change the causal mechanism in any way, then the

---

[2]We have used the subscript "i" notation in $A_i$ to denote the adjacency matrix of the intervened distribution. We will use a similar subscript notation to represent other variables related to intervened distributions

interventional distribution is equal to the observational distribution, and we would not be able to distinguish between them. This discussion suggests that it is necessary to put an additional constraint on the interventions performed. Below, we formally state the assumption that will ensure this and that it is sufficient for the identifiability of mixture distribution.

**Assumption 4.1** (Effective Intervention). *Let $\mathbb{P}_i(\boldsymbol{V}) = \mathcal{N}(\boldsymbol{m}_i, S_i)$ be an interventional distribution after intervening on node $v_i$, where $\boldsymbol{m}_i = B_i\boldsymbol{\mu_i} = B_i(\boldsymbol{\mu} + \gamma_k\boldsymbol{e}_i)$, $B_i = (I - A + \boldsymbol{e}_i\boldsymbol{c}_i^T)$ and $\boldsymbol{c}_i = (\boldsymbol{a}_i - \boldsymbol{a}'_i)$, $S_k = B_iD_iB_i^T$ and $D_i = D - \delta_i\boldsymbol{e}_i\boldsymbol{e}_i^T$ (see atomic intervention paragraph in §3). Then, at least one of the following holds: $\gamma_i \neq 0$ or $\|\boldsymbol{c}_i\| \neq 0$ or $\delta_i \neq 0$.*

Now, we are ready to state the main result of our work that will help us answer the above questions. For an exact expression of sample complexity and runtime, see Lemma 5.1.

**Theorem 4.1** (Identifiability of Mixture Parameters). *Let $\mathbb{P}_{mix}(\boldsymbol{V})$ be a mixture of soft atomic interventions defined over a Linear-SEM with additive Gaussian noise with "$n$" endogenous variables (Definition 4.1) such that the number of components $|\mathcal{I}|$ is fixed. Given Assumption 4.1 is satisfied and the causal graph corresponding to $\mathcal{M}$ doesn't violate faithfulness, then there exists an efficient algorithm that runs in time polynomial in $n$, requires $poly\left(n, \frac{1}{\epsilon}, \frac{1}{\delta}, \frac{1}{\min\left(\left\{poly(\|\boldsymbol{c}_{i_k}\|, \delta_{i_k}, \gamma_{i_k}, \frac{1}{\|A\|_F})\right\}_{i_k \in \mathcal{I}}\right)}\right)$ samples where $A$ is the adjacency matrix of underlying graph and with probability greater than $(1 - \delta)$ recovers the mixture parameters $\hat{\theta} = \{(\hat{\boldsymbol{m}}_1, \hat{S}_1, \hat{\pi}_1), \ldots, (\hat{\boldsymbol{m}}_{|\mathcal{I}|}, \hat{S}_{|\mathcal{I}|}, \hat{\pi}_{|\mathcal{I}|})\}$ such that*

$$\sum_{i_k \in \mathcal{I}} \left(\|\boldsymbol{m}_{i_k} - \hat{\boldsymbol{m}}_{\rho(i_k)}\|^2 + \|S_{i_k} - \hat{S}_{\rho(i_k)}\|^2 + |\pi_{i_k} - \hat{\pi}_{\rho(i_k)}|^2\right) \leq \epsilon^2$$

*for some permutation $\rho : \{1, 2, \ldots, |\mathcal{I}|\} \to \{1, 2, \ldots, |\mathcal{I}|\}$ and arbitrarily small $\epsilon > 0$.*

### 4.2 Causal Discovery with Mixture of Interventions

Theorem 4.1 helps us separate the mixture of interventions $\mathbb{P}_{mix}(\boldsymbol{V})$ and provides us with the parameters $\{(\boldsymbol{m}_i, S_i)\}_{i \in \mathcal{I}}$ of the distribution of all the components in the mixture. However, it does not reveal which nodes were intervened, corresponding to the different components recovered from the mixture. There has been recent progress in performing causal discovery with a disentangled set of unknown interventional distributions [26, 16]. Specifically, Squires et al. [26] proposes an algorithm (UT-IGSP) that greedily searches over the space of permutations to determine the I-MEC and the unknown intervention target of each component. UT-IGSP is consistent, i.e., it will output the correct I-MEC as the sample size goes to infinity. Thus, combining Theorem 4.1 with the UT-IGSP algorithm implies that, as sample size goes to infinity, we can recover the underlying causal graph up to its I-MEC given a mixture of interventions over a Linear-SEM with additive Gaussian noise:

**Corollary 4.1.1** (Mixture-MEC). *Given samples from a mixture of interventions $\mathcal{P}_{mix}(\boldsymbol{V})$ over a linear-SEM with additive Gaussian noise and samples from the observational distribution $\mathbb{P}(\boldsymbol{V})$, there exists a consistent algorithm that will identify the I-MEC of the underlying causal graph under the $\mathcal{I}$-faithfulness assumption (defined in Squires et al. [26]) (and restated in §A.1).*

*Proof.* The proof follows from the identifiability of the parameters of the mixture distribution (Theorem 4.1) and the consistency of UT-IGSP given by Squires et al. [26]. □

**Remark.** *The $\mathcal{I}$-faithfulness assumption imposes certain restrictions on both observational and interventional distributions. However, as noted by Squires et al. [26], in the case of Linear Gaussian distributions, the set of distributions excluded by this assumption is of measure zero. This is because the Linear Gaussian distributions, defined by a matrix $A$, that do not meet this assumption are subject to multiple polynomial constraints of the form $p(A) = 0$. It is a well-known result that for a random matrix $A$, the set of matrices that satisfy such polynomial equalities has measure zero [17].*

## 5 Proof Sketch of Theorem 4.1

Here we provide an overview of the proof of Theorem 4.1. Definition 4.1 tells us that the mixture of interventions defined over a Linear-SEM with additive Gaussian noise is a mixture of Gaussians. Learning mixtures of Gaussian is well-studied in the literature [2, 15, 7]. Since most of

these approaches require some form of separability between the distribution of components or the parameters of the distributions, in the following lemma we will first show that the covariance matrix and the mean of any interventional or observational distribution taken pairwise is *well-separated* when Assumption 4.1 holds. In particular, this *seperation* of parameters will ensure that the Gaussian mixture can be uniquely identified using the results from Belkin and Sinha [2].

**Lemma 5.1.** *[Parameter Separation] Let $\mathbb{P}_0(\boldsymbol{V})$ denote the observational distribution of a linear SEM with additive Gaussian noise "$\mathcal{M}$" (see §3) with "$n$" endogenous variables. For some $i, j \in [n] \cup \{0\}$, let $\mathbb{P}_i(\boldsymbol{V}) = \mathcal{N}(\boldsymbol{m}_i, S_i)$ and $\mathbb{P}_j(\boldsymbol{V}) = \mathcal{N}(\boldsymbol{m}_j, S_j)$ be two interventional distributions (observational if one of "$i$" or "$j$" $=0$). Then the separation between covariance $S_i$ and $S_j$ and mean $\boldsymbol{m}_i$ and $\boldsymbol{m}_j$ is lower bounded by:*

$$\|S_i - S_j\|_F^2 + \|\boldsymbol{m}_i - \boldsymbol{m}_j\|_F^2 \geq f(B, D)\Big(\|\boldsymbol{c}_i\|^2 + \|\boldsymbol{c}_j\|^2\Big) + h(B, D, \boldsymbol{\mu})\Big(\gamma_i^2 + \gamma_j^2\Big)$$
$$+ g(B)\Big(|\delta_i|\min\big(|\delta_i|, \lambda_{min}(D)\big) + |\delta_j|\min\big(|\delta_j|, \lambda_{min}(D)\big)\Big),$$

*where after intervention on node $k \in \{i, j\}$, $\|\boldsymbol{c}_k\|$ is the norm of the perturbation (or change) in the $k^{th}$ row of the adjacency matrix, $\gamma_k$ is the perturbation in the mean and $|\delta_k|$ is the perturbation in the variance of the noise distribution of node $k$ (as defined in the Atomic Intervention paragraph of §3). Also, $B = (I - A)^{-1}$, and $D$ and $\boldsymbol{\mu}$ are the covariance matrix and mean of the noise distribution in $\mathcal{M}$, respectively. Furthermore, "$f$", "$g$", and "$h$" are positive valued polynomial functions of $B$, $\boldsymbol{\mu}$ and smallest eigenvalue of $D$ (see the proof in §A.2 for the exact expressions).*

**Remark.** *If one of the distributions is observational, i.e., say $i = 0$, then the above bound holds with $\|\boldsymbol{c}_i\|^2 = 0$, $\gamma_i = 0$, and $|\delta_i| = 0$.*

**Remark.** *Different types of interventions will allow us to change certain parameters in the above bound. Let the exogenous noise variables $u_i \sim \mathcal{N}(\mu_i, \sigma_i)$ when not intervened. Now, if we intervene on node $v_i$, then the new noise variable has distribution $u_i' \sim \mathcal{N}(\mu_i + \gamma_i, \sigma_i')$ given as follows for different intervention types:*

(1) do *intervention: $\|\boldsymbol{c}_i\| \neq 0$ if $v_i$ is not a root node, $|\delta_i| \neq 0$ if $\sigma_i \neq 0$, and $\gamma_i \neq 0$ if the value of node $v_i$ is set to any value other than $\mu_i$.*
(2) stochastic *intervention: $\|\boldsymbol{c}_i\| \neq 0$ if $v_i$ is not a root node.*
(3) shift *intervention: $\gamma_i \neq 0$.*
(4) soft *intervention is the most general case, and nothing is guaranteed to be non-zero.*

Next, we restate a definition from Belkin and Sinha [2] that defines the radius of identifiability ($\mathcal{R}(\theta)$) of a probability distribution. If $\mathcal{R}(\theta) > 0$, this implies that we can uniquely identify the distribution.

**Definition 5.1.** *Let $\mathbb{P}_\theta, \theta \in \Theta$, be a family of probability distributions. For each $\theta$ we define the radius of identifiabiity $\mathcal{R}(\theta)$ as the supremum of the following set:*

$$\{r > 0 \mid \forall \theta_1 \neq \theta_2, (\|\theta_1 - \theta\| < r, \|\theta_2 - \theta\| < r) \implies (\mathbb{P}_{\theta_1} \neq \mathbb{P}_{\theta_2})\}.$$

*In other words, $\mathcal{R}(\theta)$ is the largest number, such that the open ball of radius $\mathcal{R}(\theta)$ around $\theta$ intersected with $\Theta$ is an identifiable (sub) family of the probability distribution. If no such ball exists, $\mathcal{R}(\theta) = 0$.*

Next, we restate a result from [2] adapted to our setting, which shows that there exists an efficient algorithm for disentangling a mixture of Gaussians as long as the parameters are *separated*, which will ensure that the radius of identifiability $\mathcal{R}(\theta) > 0$.

**Theorem 5.2** (Theorem 3.1 in Belkin and Sinha [2])**.** *Let $\mathbb{P}_{mix}(\boldsymbol{V})$ be a mixture of Gaussians with parameters $\theta = \{(\boldsymbol{m}_1, S_1, \pi_1), \ldots, (\boldsymbol{m}_{|\mathcal{I}|}, S_{|\mathcal{I}|}, \pi_{|\mathcal{I}|})\} \in \Theta$ where $\Theta$ is the set of parameters within a ball of radius $Q$. Then, there exists an algorithm which given $\epsilon > 0$ and $0 < \delta < 1$ and $poly\big(n, \max(\frac{1}{\epsilon}, \frac{1}{\mathcal{R}(\Theta)}), \frac{1}{\delta}, Q\big)$ samples from $\mathbb{P}_{mix}(\boldsymbol{V})$, with probability greater than $(1 - \delta)$, outputs a parameter vector $\hat{\theta} = \hat{\theta} = (\hat{\boldsymbol{m}}_1, \hat{S}_1, \hat{\pi}_1), \ldots, (\hat{\boldsymbol{m}}_{|\mathcal{I}|}, \hat{S}_{|\mathcal{I}|}, \hat{\pi}_{|\mathcal{I}|}) \in \Theta$ such that there exists a permutation $\rho : \{1, \ldots, |\mathcal{I}|\} \to \{1, \ldots, |\mathcal{I}|\}$ satisfying:*

$$\sum_{i_k \in \mathcal{I}} \Big(\|\boldsymbol{m}_{i_k} - \hat{\boldsymbol{m}}_{\rho(i_k)}\|^2 + \|S_{i_k} - \hat{S}_{\rho(i_k)}\|^2 + |\pi_{i_k} - \hat{\pi}_{\rho(i_k)}|^2\Big) \leq \epsilon^2,$$

*where the radius of identifiability $\mathcal{R}(\theta)$ is lower bounded by:*

$$\big(\mathcal{R}(\theta)\big)^2 \geq \min\Big(\frac{1}{4}\min_{i \neq j}(\|\boldsymbol{m}_i - \boldsymbol{m}_j\|^2 + \|S_i - S_j\|^2), \min_i \pi_i\Big).$$

**Algorithm 1:** Mixture-UTIGSP

---

**input** : mixed dataset $(\mathcal{D}_{\text{mix}})$, observational data $(\mathcal{D}_{\text{obs}})$, number of nodes $(n)$, cutoff ratio $(\tau)$

**output**: Mixture Distribution Parameters $(\theta)$, Intervention Targets $(\mathcal{I})$, causal graph $(\hat{\mathcal{G}})$

1. Esstimate $\theta_k \triangleq \left\{ (\hat{\boldsymbol{\mu}}_1, \hat{S}_1), \ldots (\hat{\boldsymbol{\mu}}_k, \hat{S}_k) \right\} = GaussianMixtureModel(\mathcal{D}_{mix}, k)$ for each possible number of component in the mixture i.e $k \in [n+1]$. Define $\Theta \triangleq \{\theta_1, \ldots, \theta_{n+1}\}$ be the set of estimated parameters and $\mathcal{L} = \{l_1, \ldots, l_{n+1}\}$ be the log-likelihood of the mixture data corresponding to the models with a different number of components.

2. To estimate the number of components in the mixture $(k_*)$ iterate over $k = (n+1)$ to 2:

   (a) stop where the relative change in the likelihood increases above a cutoff ratio i.e $\frac{|l_k - l_{k-1}|}{l_k} > \tau$

   (b) $k_* = k$ if the stopping criteria is met otherwise $k_* = 1$.

3. $\mathcal{I}, \hat{\mathcal{G}} = \text{UT-IGSP}(\mathcal{D}_{\text{obs}}, \theta_{k_*})$ [26]

**return** $\Theta, \mathcal{I}, \hat{\mathcal{G}}$

---

Our Theorem 4.1 along with Assumption 4.1 states that for every pair $i, j \in ([n] \cup \{0\})^{\otimes 2}$ we have $\|S_i - S_j\|_F^2 + \|\boldsymbol{m}_i - \boldsymbol{m}_j\|_F^2 > 0$. Also, by construction of the mixture of interventions (in Definition 4.1), we have $\pi_i > 0, \forall i \in [n] \cup \{0\}$. This implies that the radius of convergence $\mathcal{R}(\theta) > 0$ and thus the parameters of the mixture of interventions $\mathbb{P}(\boldsymbol{V})$ can be identified uniquely given samples from the mixture distribution with sample size inversely proportional to $\mathcal{R}(\theta)$.

## 6 Empirical Results

### 6.1 Experiment on Simulated Datasets

Proposition 4.1.1 establishes that given samples from the mixture distribution, one can identify the underlying causal graph up to its I-MEC. To learn the causal graph, we first disentangle the mixture. Theorem 4.1 and 5.2 show that the sample complexity of our mixture disentangling algorithm is inversely proportional to various parameters of the underlying system and the intervention parameters $(\gamma, |\delta|$ and $\|\boldsymbol{c}\|)$. Our simulation study further validates our theoretical results and characterizes the end-to-end performance of identifying the causal graph with such mixture data and its dependence on the above-mentioned parameters.

**Simulation Setup** We consider data generated from a Linear-SEM with additive Gaussian noise, $\boldsymbol{x} = (I - A)^{-1} \boldsymbol{\eta}$ (see §3), with n endogenous variables and corresponding exogenous (noise) variables. Here $\boldsymbol{\eta} \sim \mathcal{N}(0, D)$, where the noise covariance matrix is diagonal with entries $D = diag(\sigma^2, \ldots, \sigma^2)$ and $\sigma = 1$ unless otherwise specified. $A$ is the (lower-triangular) weighted adjacency matrix whose weights are sampled in the range $[-1, -0.5] \cup [0.5, 1]$ bounded away from 0. Let $\mathcal{G}^*$ denote the causal graph corresponding to this linear SEM with edge $i \to j \Leftrightarrow A_{ji} > 0$. By sampling from the resulting multivariate Gaussian distribution, we obtain *observational* data. Next, for each causal graph, we generate separate *interventional* data by intervening on a given set of nodes in the graph one at a time (atomic interventions), which is again a Gaussian distribution but with different parameters (see Atomic Intervention paragraph in §3). We experiment with two settings:

   (1) *all*: where we perform an atomic intervention on all nodes in the graph.

   (2) *half*: where we perform an atomic intervention on a randomly selected half of the nodes.

Then, the mixed data is generated by pooling all the individual atomic interventions and observational data with equal proportions into a single dataset. The decision to use equal proportions of samples from all components is solely intended to simplify the design choices for the experiment setup. In our experiment, we vary the total number of samples in the mixed dataset as $N \in \{2^{10}, 2^{11}, \ldots, 2^{17}\}$. In particular, we perform two kinds of atomic interventions: "*do*" and "stochastic" (see Interventions paragraph in §3 for a formal definition). The initial noise distribution for all the nodes is univariate Gaussian distribution $\mathcal{N}(0, 1)$. In our experiments for *do* interventions, instead of setting the final variance of noise distribution to 0, we set it to a very small value of $10^{-9}$ for numerical stability. Unless otherwise specified, we perform 10 runs for each experimental setting and plot the 0.05 and 0.95 quantiles. See B for additional details on the experimental setup.

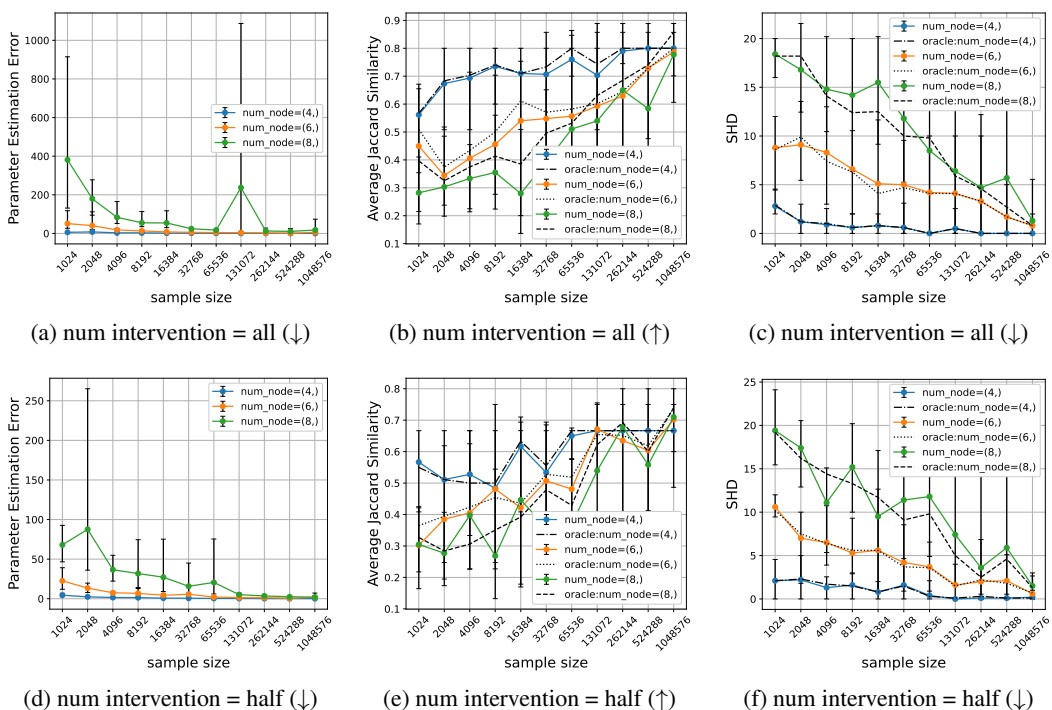

| (a) num intervention = all (↓) | (b) num intervention = all (↑) | (c) num intervention = all (↓) |
| (d) num intervention = half (↓) | (e) num intervention = half (↑) | (f) num intervention = half (↓) |

Figure 1: **Performance of Alg. 1 as we vary sample size and number of nodes**: The first row (a-c) shows the performance when the mixed data contains atomic intervention on all the nodes and observational data. The second row (d-f) shows the performance when the number of atomic interventions (chosen randomly) in mixed data is taken to be half of the number of nodes along with observational data. The column shows different evaluation metrics, i.e., Parameter Estimation Error, Average Jaccard Similarity, and SHD. The symbols (↑) represent higher is better, and (↑) represents the opposite (see Evaluation metric paragraph in §6). In summary, performance improves for both cases as the number of samples increases. However, the graph with more nodes requires a larger sample to perform similarly. For a detailed discussion, see §6.1.

**Method Description and Evaluation Metrics**   Given the mixed data generated from the underlying true causal graph $\mathcal{G}^*$, the goal is to estimate the underlying causal graph $\hat{\mathcal{G}}$. We break down the task into two steps. First, we disentangle the mixture data and identify the parameters of the individual interventional and/or observational distributions. Our theoretical result (Theorem 4.1) uses [2] for identifiability of the parameters of a mixture of Gaussians. Since they only show the existence of such an algorithm, we use the standard sklearn python package [19] that implements an EM algorithm to estimate the parameters of the mixture. Importantly, our experiment doesn't require prior knowledge about the number of components (k) in the mixture. We train separate mixture models varying the number of components. Then we select the optimal number of components using the log-likelihood curve of the fitted Gaussian mixture model using a simple thresholding heuristic (see step 2 in Alg. 1). We leave the exploration of better model selection criteria for future work. For all our simulation experiments, unless otherwise specified, we use a cutoff threshold of 0.07, chosen arbitrarily. In §B, we experiment with different values of this threshold and show that Mixture-UTIGSP is robust to this choice. The intervention targets present in the mixture are still unknown at this step. Next, we provide the estimated mixture parameters to an existing causal discovery algorithm with unknown intervention targets (UT-IGSP [26]), which internally estimates the unknown intervention targets and outputs one of the possible graphs from the estimated I-MEC. We assume that observational data is given as an input to the UT-IGSP algorithm. The proposed algorithm is provided in Alg. 1. See B for our hyperparameter choice and other experimental details.

**Evaluation Metrics**   We evaluate the performance of Mixture-UTIGSP (Alg. 1) on three metrics:

*Parameter Estimation Error:* This metric measures the least absolute error between the estimated parameters (mean and covariance matrix defining each individual distribution) after the first step

Table 1: **Performance of Alg. 1 on Protein Signalling Dataset [22]**: We evaluate the performance of Mixture-UTIGSP as we vary the cutoff ratio to select the number of component in the mixture. The second column shows the number of estimated components where the actual number of components in the mixture is 6. The third and fourth columns show the Jaccard Similarity of the identified intervention target of Mixture-UTIGSP and oracle versions of the UT-IGSP algorithm. The fourth and last column shows the SHD between the estimated and true causal graphs for both methods respectively. Overall we observe that at a lower cutoff threshold Mixture-UTIGSP is able to perform as well as the oracle UT-IGSP algorithm on all the metrics. See §B.2 for detailed discussion.

| cutoff ratio | Estimated/True Component | JS | Oracle JS | SHD | Oracle SHD |
|---|---|---|---|---|---|
| 0.01 | 4/6 | 0.07 | 0.05 | 15 | 19 |
| 0.07 | 2/6 | 0.04 | 0.05 | 18 | 17 |
| 0.15 | 1/6 | 0.00 | 0.05 | 16 | 18 |
| 0.30 | 1/6 | 0.00 | 0.05 | 18 | 17 |
| avg | 2/6 | 0.03 | 0.05 | 16.75 | 17.75 |

of Mixture-UTIGSP matched with the ground truth parameters considering all possible matchings between the components averaged over all runs. See §B.5 for details.

*Average Jaccard Similarity (JS):* We measure the average Jaccard similarity between the estimated intervention target and the corresponding ground truth (atomic) intervention target. We use the matching between the estimated and ground truth components found while calculating the parameter estimation error averaged over all runs. See §B.5 for details.

*Average Structural Hamming Distance (SHD):* Given the estimated and ground truth graphs, we compute the SHD between the two graphs averaged over all runs.

**Results** Fig. 1 shows the performance of Alg. 1 in the two settings, *"all"* in the first row and *"half"* in the second row (see Simulation setup above). The first column shows the performance of the first step of Alg. 1 where the mixture parameters are identified. We observe that parameter estimation error decreases as the number of samples increases in both settings. As expected, larger graphs require a larger sample size to perform similarly to smaller-sized graphs within each setting.

Step 1 of our Alg. 1 only recovers the parameters ($\{(\boldsymbol{m}_i, S_i)\}_{i=1}^k$) of the components present in the mixture distribution. In step 2 of our Alg. 1, we call UT-IGSP [26] that identifies the individual intervention targets from the estimated distribution parameters and also returns a causal graph from the estimated I-MEC. Fig. 1b and 1e show the average Jaccard Similarity between the ground truth and the estimated intervention targets. The colored lines denote experiments on graphs with different numbers of nodes. The corresponding dotted lines show the oracle performance of UT-IGSP when the separated ground truth mixture distributions were given as input. As expected, in both the settings (Fig. 1b and 1e), the oracle version performs much better compared to its non-oracle counterpart for small sample sizes ($2^{10}$ to $2^{14}$) but performs similarly as sample size increases.

Finally, in the third column (Fig. 1c and 1f), we calculate the SHD between the estimated causal graph $\hat{\mathcal{G}}$ and the ground truth causal graph $\mathcal{G}^*$. The SHD of the graph estimated by Mixture-UTIGSP and the oracle version are similar for different node settings and all sample sizes. This suggests that small errors in the estimation of the parameters of the mixture distribution don't affect the estimation of the underlying causal graph.

**Additional Experiments:** In §B, we provide details on the experimental setup and additional results. In Fig. 2, we plot two additional metrics for the simulation experiments. The first metric is the number of estimated components in the mixture and the second metric is the the error in estimation of the mixing coefficient. In Fig. 3, we study the sensitivity of the cutoff ratio used by Mixture-UTIGSP to select the number of components in the mixture. Next, in Fig. 4, we evaluate the performance of our Alg. 1 as we vary the density, i.e., the expected number of edges in the graph. In Fig. 6, we show how the sparsity of the graph and other intervention parameters like the value of the new mean and variance of the noise distribution after intervention affects the performance of Alg. 1.

## 6.2 Experiment on Biological Dataset

We evaluate our method on the Protein Signaling dataset [22] to demonstrate real-world applicability. The dataset is collected from flow cytometry measurement of 11 phosphorylated proteins and phospholipids and is widely used in causal discovery literature [31, 26]. The dataset consists of 5846 measurements with different experimental conditions and perturbations. Following Wang et al. [31], we define the subset of the dataset as observational, where only the receptor enzymes were perturbed in the experiment. Next, we select other 5 subsets of the dataset where a signaling protein is also perturbed in addition to the receptor enzyme. The observational dataset consists of 1755 samples, and the 5 interventional datasets have 911, 723, 810, 799, and 848 samples, respectively. The mixed dataset is created by merging all the observational and interventional datasets.

The total number of nodes in the underlying causal graph is 11. Thus, the maximum number of possible components in the mixture is 12 (11 single-node interventional distribution and one observational). In the mixture dataset described above, we have 6 components (1 observational and 5 interventional). The second column in Table 1 shows that Mixture-UTIGSP recovers 4 components close to the ground truth 6 when the cutoff ratio is 0.01 (step 2 of Alg. 1). Next, we give the disentangled dataset from the first step of our algorithm to identify the unknown target. Though the Jaccard similarity of the recovered target is not very high (average of 0.03 shown in the last row of the third column, where the maximum value is 1.0), it is similar to that of the oracle performance of UT-IGSP when the disentangled ground truth mixture distributions were given as input. This shows that it is difficult to identify the correct intervention targets even with correctly disentangled data. Also, the SHD between the recovered graph and the widely accepted ground truth graph for Mixture-UTIGSP (ours) and UT-IGSP (oracle) is very close. Overall, at a lower cutoff ratio, the performance of Mixture-UTIGSP is close to the Oracle UT-IGSP algorithm. Unlike the simulation case (see Fig. 3), Mixture-UTIGSP's performance is sensitive to the choice of the cutoff ratio on this dataset. In Fig. 5, we plot the ground truth graph curated by the domain expert alongside the estimated causal graph for visualization.

## 7 Conclusion

We studied the problem of learning the mixture distribution generated from observational and/or multiple unknown interventional distributions generated from the underlying causal graph. We show that the parameters of the mixture distribution can be uniquely identified under the mild assumption that ensures that any intervention changes the distribution of the observed variables. As a consequence of our identifiability result, under an interventional faithfulness assumption (Squires et al. [26]), we show that the underlying true causal graph can be identified up to its I-MEC based on a mixture of unknown interventions, thereby obtaining the same identifiability results as in the unmixed setting. Finally, we conduct a simulation study to validate our findings empirically. We demonstrate that as the sample size increases, we obtain parameter estimates of the mixture distribution that are closer to the ground truth and, as a result, we eventually recover the correct underlying causal graph.

## 8 Limitations and Future Work

Since our work is the first to study the problem of a mixture of causal interventions without assuming knowledge of causal graphs, we have restricted our attention to one particular family of causal models—Linear-SEM with additive Gaussian noise. In the future, it would be interesting to study this problem for a more general family of causal models. Further, our work uses Belkin and Sinha [2] for identifying the parameters of a mixture of Gaussians, which assumes that the number of components in the mixture is fixed. Recent progress in [7] gives an efficient algorithm for recovering the parameters when the number of components is almost $\sqrt{n}$, where $n$ is the number of variables. However, they only work in perturbative settings, and proving the result for non-perturbative settings is out of the scope of the current paper. Finally, to identify the parameters of the mixture distribution in our empirical study, we use heuristics to estimate the number of components. It would be interesting to explore other methods to automatically select the number of components.

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

## A Missing Proofs

### A.1 I-faithfulness in UT-IGSP algorithm.

Here, we restate the assumption from Squires et al. [26] that is needed for consistency of the UT-IGSP algorithm:

**Assumption A.1** ($\mathcal{I}$-faithfulness assumption). *Let $\mathcal{I}$ be a list of intervention targets. The set of distributions $\{f^{obs}\} \cup \{f^I\}_{I_{\mathcal{I}}}$ is $\mathcal{I}$-faithful with respect to a DAG $\mathcal{G}$ if $f^{obs}$ is faithful with respect to $\mathcal{G}$ and for any $I^k \in \mathcal{I}$ and disjoint $A, C \subseteq [p]$, we have that $(A \perp\!\!\!\perp \zeta_k | C \cup \zeta_{\mathcal{I}\setminus\{k\}})_{\mathcal{G}^{\mathcal{I}}}$ if and only if $f^k(x_A|x_C) = f^{obs}(x_A|x_C)$.*

### A.2 Proof of Lemma 5.1

**Lemma 5.1.** *[Parameter Separation] Let $\mathbb{P}_0(\boldsymbol{V})$ denote the observational distribution of a linear SEM with additive Gaussian noise "$\mathcal{M}$" (see §3) with "n" endogenous variables. For some $i, j \in [n] \cup \{0\}$, let $\mathbb{P}_i(\boldsymbol{V}) = \mathcal{N}(\boldsymbol{m}_i, S_i)$ and $\mathbb{P}_j(\boldsymbol{V}) = \mathcal{N}(\boldsymbol{m}_j, S_j)$ be two interventional distributions (observational if one of "i" or "j" =0). Then the separation between covariance $S_i$ and $S_j$ and mean $\boldsymbol{m}_i$ and $\boldsymbol{m}_j$ is lower bounded by:*

$$\|S_i - S_j\|_F^2 + \|\boldsymbol{m}_i - \boldsymbol{m}_j\|_F^2 \geq f(B, D)\Big(\|\boldsymbol{c}_i\|^2 + \|\boldsymbol{c}_j\|^2\Big) + h(B, D, \boldsymbol{\mu})\Big(\gamma_i^2 + \gamma_j^2\Big)$$
$$+ g(B)\Big(|\delta_i| \min\big(|\delta_i|, \lambda_{min}(D)\big) + |\delta_j| \min\big(|\delta_j|, \lambda_{min}(D)\big)\Big),$$

*where after intervention on node $k \in \{i, j\}$, $\|\boldsymbol{c}_k\|$ is the norm of the perturbation (or change) in the $k^{th}$ row of the adjacency matrix, $\gamma_k$ is the perturbation in the mean and $|\delta_k|$ is the perturbation in the variance of the noise distribution of node $k$ (as defined in the Atomic Intervention paragraph of §3). Also, $B = (I - A)^{-1}$, and $D$ and $\boldsymbol{\mu}$ are the covariance matrix and mean of the noise distribution in $\mathcal{M}$, respectively. Furthermore, "f", "g", and "h" are positive valued polynomial functions of $B$, $\boldsymbol{\mu}$ and smallest eigenvalue of $D$ (see the proof in §A.2 for the exact expressions).*

*Proof.* First, we state a lemma that will give us a lower bound on the separation between the covariance matrix of two intervention distributions (or one could be observational). The proof is given in §A.3.

**Lemma A.1** (Minimum Covariance Separation). *Let $S_i$ and $S_j$ be the covariance matrix of the Gaussian distribution corresponding to intervention on node $V_i$ and $V_j$ in the causal graph. Let, without loss of generality, $V_i$ be topologically greater than $V_j$ in the causal graph. Let $B = (I - A)^{-1}$, $D = diag(\sigma_1, \ldots, \sigma_n)$ be the covariance matrix of the noise distribution, let $S = BDB^T$ be the covariance matrix of the observed distribution $P(\boldsymbol{V})$, and let $\boldsymbol{c}_i = \boldsymbol{a}_i - \boldsymbol{a}_i'$ be the soft intervention performed on node $V_i$ (see §3 for definitions). Then we have:*

$$\|S_i - S_j\|_F^2 \geq f(B, D)\Big(\|\boldsymbol{c}_i\|^2 + \|\boldsymbol{c}_j\|^2\Big)$$
$$+ g(B)\Big(|\delta_i| \min\big(|\delta_i|, \lambda_{min}(D)\big) + |\delta_j| \min\big(|\delta_j|, \lambda_{min}(D)\big)\Big).$$

*If one of the covariance matrices is from the observational distribution, i.e., say $S_j = S$ then the above lower bounds still holds with $\|\boldsymbol{c}_j\| = 0$ and $\delta_j = 0$.*

Using the above Lemma A.1 and substituting the explicit form of $f(B, D)$ from Eq. 19 we obtain:

$$\|S_i - S_j\|_F^2 \geq f(B, D)\Big(\|\boldsymbol{c}_i\|^2 + \|\boldsymbol{c}_j\|^2\Big)$$
$$+ \underbrace{g(B)\Big(|\delta_i| \min\big(|\delta_i|, \lambda_{min}(D)\big) + |\delta_j| \min\big(|\delta_j|, \lambda_{min}(D)\big)\Big)}_{\triangleq \omega} \tag{5}$$
$$\geq \underbrace{\frac{f(B, D)}{2}\Big(\|\boldsymbol{c}_i\|^2 + \|\boldsymbol{c}_j\|^2\Big) + \omega}_{\triangleq \zeta} + \frac{\lambda_{min}^2(D)\Big(\|\boldsymbol{c}_i\|^2 + \|\boldsymbol{c}_j\|^2\Big)}{8\|B^{-1}\|_F^4}.$$

Next, we state another lemma that will give us a lower bound on the separation of the mean of two interventional distributions (or one of them could be observational). The proof of the lemma below is given in §A.5.

**Lemma A.2** (Minimum Mean Separation). *Let $\boldsymbol{m}_i$ and $\boldsymbol{m}_j$ denote the mean of the Gaussian distribution corresponding to the intervention on node $V_i$ and $V_j$ in the causal graph. Then we have:*

$$
\|\boldsymbol{m}_i - \boldsymbol{m}_j\|_F^2 \geq
\begin{cases}
\frac{\gamma_i^2}{4\|B^{-1}\|_F^2} + \frac{\gamma_j^2}{4\|B^{-1}\|_F^2}, & \psi_i^+, \psi_j^+ \text{ are active} \\
\frac{\gamma_i^2}{4\|B^{-1}\|_F^2}, & \psi_i^+, \psi_j^- \text{ are active and } \frac{\gamma_j^2}{4\|B\boldsymbol{\mu}\|^2} \leq \|\boldsymbol{c}_j\|^2 \\
\frac{\gamma_j^2}{4\|B^{-1}\|_F^2}, & \psi_i^-, \psi_j^+ \text{ are active and } \frac{\gamma_i^2}{4\|B\boldsymbol{\mu}\|^2} \leq \|\boldsymbol{c}_i\|^2 \\
0, & \psi_i^-, \psi_j^- \text{ are active and } \frac{\gamma_i^2}{4\|B\boldsymbol{\mu}\|^2} \leq \|\boldsymbol{c}_i\|^2, \frac{\gamma_j^2}{4\|B\boldsymbol{\mu}\|^2} \leq \|\boldsymbol{c}_j\|^2
\end{cases}
$$

*where $\psi_i^+ \triangleq \left(|\boldsymbol{c}_i^T B\boldsymbol{\mu}| < \frac{|\gamma_i|}{2} \text{ or } |\boldsymbol{c}_i^T B\boldsymbol{\mu}| > \frac{3|\gamma_i|}{2}\right)$, $\psi_i^- \triangleq \frac{|\gamma_i|}{2} \leq |\boldsymbol{c}_i^T B\boldsymbol{\mu}| \leq \frac{3|\gamma_i|}{2}$ and similarly for $\psi_j^+$ and $\psi_j^-$. If one of the means say $\boldsymbol{m}_j$ is from the observational distribution, i.e., $\boldsymbol{m}_j = \boldsymbol{m} = B\boldsymbol{\mu}$, then setting $\gamma_j = 0$ above will give the appropriate bounds and only case 2 and 4 are applicable.*

Now, From Case 1 of the above Lemma A.2 (Eq. 42, $\psi_i^+, \psi_j^+$ is active) and above equation we have:

$$
\|S_i - S_j\|_F^2 + \|\boldsymbol{m}_i - \boldsymbol{m}_j\|_F^2 \geq \zeta + \frac{\lambda_{min}^2(D)\left(\|\boldsymbol{c}_i\|^2 + \|\boldsymbol{c}_j\|^2\right)}{8\|B^{-1}\|_F^4} + \frac{\gamma_i^2}{4\|B^{-1}\|_F^2} + \frac{\gamma_j^2}{4\|B^{-1}\|_F^2}.
\tag{6}
$$

From Case 2 of the above Lemma A.2 (Eq. 42, $\psi_i^+, \psi_j^-$ is active) and using $\frac{\gamma_j^2}{4\|B\boldsymbol{\mu}\|^2} \leq \|\boldsymbol{c}_j\|^2$ we have:

$$
\|S_i - S_j\|_F^2 + \|\boldsymbol{m}_i - \boldsymbol{m}_j\|_F^2 \geq \zeta + \frac{\lambda_{min}^2(D)\|\boldsymbol{c}_i\|^2}{8\|B^{-1}\|_F^4} + \frac{\gamma_i^2}{4\|B^{-1}\|_F^2} + \frac{\lambda_{min}^2(D)\gamma_j^2}{32\|B^{-1}\|_F^4\|B\boldsymbol{\mu}\|^2}.
\tag{7}
$$

From Case 3 of the above Lemma A.2 (Eq. 42, $\psi_i^-, \psi_j^+$ is active) and using $\frac{\gamma_i^2}{4\|B\boldsymbol{\mu}\|^2} \leq \|\boldsymbol{c}_i\|^2$ we have:

$$
\|S_i - S_j\|_F^2 + \|\boldsymbol{m}_i - \boldsymbol{m}_j\|_F^2 \geq \zeta + \frac{\lambda_{min}^2(D)\|\boldsymbol{c}_j\|^2}{8\|B^{-1}\|_F^4} + \frac{\gamma_j^2}{4\|B^{-1}\|_F^2} + \frac{\lambda_{min}^2(D)\gamma_i^2}{32\|B^{-1}\|_F^4\|B\boldsymbol{\mu}\|^2}.
\tag{8}
$$

Similarly, for Case 4 and using $\frac{\gamma_i^2}{4\|B\boldsymbol{\mu}\|^2} \leq \|\boldsymbol{c}_i\|^2$ and $\frac{\gamma_j^2}{4\|B\boldsymbol{\mu}\|^2} \leq \|\boldsymbol{c}_j\|^2$ we have:

$$
\|S_i - S_j\|_F^2 + \|\boldsymbol{m}_i - \boldsymbol{m}_j\|_F^2 \geq \zeta + \frac{\lambda_{min}^2(D)\gamma_i^2}{32\|B^{-1}\|_F^4\|B\boldsymbol{\mu}\|^2} + \frac{\lambda_{min}^2(D)\gamma_j^2}{32\|B^{-1}\|_F^4\|B\boldsymbol{\mu}\|^2}.
\tag{9}
$$

Next, combining Eq. 6, 7, 8, 9 we have:

$$
\begin{aligned}
\|S_i - S_j\|_F^2 + \|\boldsymbol{m}_i - \boldsymbol{m}_j\|_F^2 &\geq \zeta + \frac{\gamma_i^2 + \gamma_j^2}{\max\left(4\|B^{-1}\|_F^2, \frac{32\|B^{-1}\|_F^4\|B\boldsymbol{\mu}\|^2}{\lambda_{min}^2(D)}\right)} \\
&\geq \frac{f(B, D)}{2}\left(\|\boldsymbol{c}_i\|^2 + \|\boldsymbol{c}_j\|^2\right) + g(B, D)\left(|\delta_i| + |\delta_j|\right) \\
&\quad + h(B, D, \boldsymbol{\mu})(\gamma_i^2 + \gamma_j^2).
\end{aligned}
\tag{10}
$$

For the case when one of the distributions is observational, say w.l.o.g. $S_j = S$ and $\boldsymbol{m}_j = \boldsymbol{m}$, then the same analysis holds since $\boldsymbol{c}_j = 0$ and $\delta_j = 0$ (from Lemma A.1) and only the analyses of case 2 and case 4 are applicable (from Lemma A.2), thereby completing the proof. □

## A.3  Proof of Lemma A.1

*Proof.* Without loss of generality, throughout our analysis, we will assume that the endogenous variables $V_1, \ldots, V_n$ are topologically ordered based on the underlying causal graph. Thus, the

corresponding adjacency matrix $A$ is lower triangular. The value of $\|S_i - S_j\|_F^2$ will be unaffected by any permutation of the node order in the matrix $A$ as shown next. Let $\tilde{A} = PAP^T$ be the adjacency matrix when the nodes are permuted by the permutation matrix $P$ where $PP^T = I$. Also, let the corresponding permuted covariance matrix be $\tilde{S} = \tilde{B}\tilde{D}\tilde{B}^T$ where $\tilde{B} = (I - \tilde{A})^{-1} = P(I - A)^{-1}P^T = PBP^T \implies \tilde{S} = PBP^T PDP^T PB^T P^T = PBDB^T P^T = PSP^T$. Similarly, we have $\tilde{S}_i = PS_iP^T$ and $\tilde{S}_j = PS_jP^T$. Now we have:

$$\|\tilde{S}_i - \tilde{S}_j\|_F^2 = \|P(S_i - S_j)P^T\|_F^2 = \|S_i - S_j\|_F^2, \tag{11}$$

since the permutation matrix only permutes the row and column in the above equation, the Frobenius norm remains the same.

First, we state a lemma that characterizes the covariance matrix $S_i$ of the interventional distribution. The proof of this lemma can be found in §A.4.

**Lemma A.3** (Covariance Matrix Update). *Let $\mathbb{P}_i(\boldsymbol{V}) = \mathcal{N}(\boldsymbol{m}_i, S_i)$ be an interventional distribution and let the endogenous nodes $\boldsymbol{V}_1, \ldots \boldsymbol{V}_n$ be topologically ordered based on the underlying causal graph, then we have:*

$$S_i = \begin{cases} S - \delta_i \boldsymbol{r}_i \boldsymbol{r}_i^T, & \text{for root node} \\ B_i D B_i^T - \delta_i \boldsymbol{r}_i \boldsymbol{r}_i^T, & \text{otherwise} \end{cases}$$

*where $\boldsymbol{r}_i = B\boldsymbol{e}_i$, $\delta_i = \sigma_i - \sigma_i'$, $B_i = B - B\boldsymbol{e}_i \boldsymbol{c}_i^T B$, $B = (I - A)^{-1}$, $S$ is the covariance matrix of the observational distribution and $D$ is the covariance matrix of the observational noise distribution (see Atomic Intervention paragraph in §3).*

Thus from the above Lemma A.3, we have $S_i = B_i D B_i^T - \delta_i \boldsymbol{v_i} \boldsymbol{v_i}^T$. Substituting the value of $B_i$ from the above lemma again we get:

$$\begin{aligned} S_i &= BDB^T - BD\boldsymbol{u}_i \boldsymbol{v}_i^T - \boldsymbol{v}_i \boldsymbol{u}_i^T DB^T + \eta_i \boldsymbol{v}_i \boldsymbol{v}_i^T - \delta_i \boldsymbol{v}_i \boldsymbol{v}_i^T \\ &= S - BDB^T \boldsymbol{c}_i \boldsymbol{e}_i^T B^T - B\boldsymbol{e}_i \boldsymbol{c}_i^T BDB^T + \eta_i B\boldsymbol{e}_i \boldsymbol{e}_i^T B^T - \delta_i B\boldsymbol{e}_i \boldsymbol{e}_i^T B^T, \end{aligned} \tag{12}$$

where $\boldsymbol{u}_i = B^T \boldsymbol{c}_i$, $\boldsymbol{v}_i = B\boldsymbol{e}_i$, $\eta_i = \boldsymbol{u}_i^T D\boldsymbol{u}_i = \boldsymbol{c}_i^T S\boldsymbol{c}_i$. Next, multiplying the LHS and RHS of the above equation with $B^{-1}$ and $B^{-T}$ we get:

$$B^{-1}(S_i - S)B^{-T} = \eta_i \boldsymbol{e}_i \boldsymbol{e}_i^T - DB^T \boldsymbol{c}_i \boldsymbol{e}_i^T - \boldsymbol{e}_i \boldsymbol{c}_i^T BD - \delta_i \boldsymbol{e}_i \boldsymbol{e}_i^T. \tag{13}$$

The absolute value of $(i, k)^{\text{th}}$ ($k \in [n]$) entry of this matrix is given by:

$$\boldsymbol{e}_i^T B^{-1}(S_i - S)B^{-T}\boldsymbol{e}_k = \begin{cases} -D_{kk}\boldsymbol{c}_i^T B\boldsymbol{e}_k, & k \neq i \\ \eta_i - 2D_{ii}\boldsymbol{c}_i^T B\boldsymbol{e}_i^{\,0} - \delta_i = \eta_i - \delta_i, & k = i \end{cases} \tag{14}$$

where in the second case $\boldsymbol{c}_i^T B\boldsymbol{e}_i = 0$ and $D_{kk} = \sigma_k$ is the $k^{\text{th}}$ diagonal entry of the matrix $D$. Similarly, the $(j, k)^{\text{th}}$ entry of this matrix is given by:

$$\boldsymbol{e}_j^T B^{-1}(S_i - S)B^{-T}\boldsymbol{e}_k = \begin{cases} 0, & k \neq i \\ -D_{jj}\boldsymbol{e}_j^T B^T \boldsymbol{c}_i & k = i \end{cases} \tag{15}$$

Similarly, the covariance matrix of node $V_j$ and $j \neq i$ is given by:

$$S_j = S - BDB^T \boldsymbol{c}_j \boldsymbol{e}_j^T B^T - B\boldsymbol{e}_j \boldsymbol{c}_j^T BDB^T + \eta_j B\boldsymbol{e}_j \boldsymbol{e}_j^T B^T - \delta_j B\boldsymbol{e}_j \boldsymbol{e}_j^T B^T$$
$$B^{-1}(S_j - S)B^{-T} = \eta_j \boldsymbol{e}_j \boldsymbol{e}_j^T - DB^T \boldsymbol{c}_j \boldsymbol{e}_j^T - \boldsymbol{e}_j \boldsymbol{c}_j^T BD - \delta_j \boldsymbol{e}_j \boldsymbol{e}_j^T. \tag{16}$$

Thus the absolute value of the $(i, k)^{\text{th}}$ entry of the above matrix is given by:

$$|\boldsymbol{e}_i^T B^{-1}(S_j - S)B^{-T}\boldsymbol{e}_k| = |D_{ii}\boldsymbol{e}_i B^T \boldsymbol{c}_j(\boldsymbol{e}_j^T \boldsymbol{e}_k)| = 0, \tag{17}$$

since we had assumed without loss of generality that $V_j \prec V_i \implies \boldsymbol{e}_i B^T \boldsymbol{c}_j = 0$. Similarly, the $(j, k)^{\text{th}}$ entry of this matrix is given by:

$$\boldsymbol{e}_j^T B^{-1}(S_j - S)B^{-T}\boldsymbol{e}_k = \begin{cases} -D_{kk}\boldsymbol{c}_j^T B\boldsymbol{e}_k, & k \neq j \\ \eta_j - 2D_{jj}\boldsymbol{e}_j^T B^T \boldsymbol{c}_j - \delta_j = \eta_j - \delta_j & k = j \end{cases} \tag{18}$$

where in the second case $e_j^T B^T \boldsymbol{c}_j = 0$. Thus, taking the difference of the $i^{\text{th}}$ and $j^{\text{th}}$ row of both the matrix (from Eq. 14, 15, 17 and 18 and using the fact that $\boldsymbol{c}_i^T B \boldsymbol{e}_i = 0$, $\boldsymbol{c}_j^T B \boldsymbol{e}_j = 0$ and $\boldsymbol{c}_j^T B \boldsymbol{e}_i = 0$ since without loss of generality we have $V_j \prec V_i$) we obtain the following lower bound:

$$
\begin{aligned}
\|B^{-1}(S_i - S_j)B^{-T}\|_F^2 &\geq \sum_k (e_i B^{-1}(S_i - S_j)B^{-T} e_k)^2 + \sum_k (e_j B^{-1}(S_i - S_j)B^{-T} e_k)^2 \\
&\geq \left[ \sum_{k=1}^n (D_{kk}\boldsymbol{c}_i^T B \boldsymbol{e}_k)^2 + (\eta_i - \delta_i)^2 \right] \\
&\quad + \left[ \sum_{k \neq i} (D_{kk}\boldsymbol{c}_j^T B \boldsymbol{e}_k)^2 + (D_{jj}\boldsymbol{e}_j^T B^T \boldsymbol{c}_i - \cancelto{0}{D_{ii}\boldsymbol{e}_j^T B \boldsymbol{e}_i})^2 + (\eta_j - \delta_j)^2 \right] \\
&= \sum_{k=1}^n D_{kk}^2 \boldsymbol{c}_i^T B \boldsymbol{e}_k \boldsymbol{e}_k^T B^T \boldsymbol{c}_i + (\eta_i - \delta_i)^2 \\
&\quad + \sum_{k=1}^n D_{kk}^2 \boldsymbol{c}_j^T B \boldsymbol{e}_k \boldsymbol{e}_k^T B^T \boldsymbol{c}_j + (\eta_j - \delta_j)^2 + (D_{jj}\boldsymbol{e}_j^T B^T \boldsymbol{c}_i)^2 \\
&\geq \lambda_{min}(D)\boldsymbol{c}_i^T B \Big( \sum_{k=1}^n D_{kk}\boldsymbol{e}_k \boldsymbol{e}_k^T \Big) B^T \boldsymbol{c}_i + (\eta_i - \delta_i)^2 \\
&\quad + \lambda_{min}(D)\boldsymbol{c}_j^T B \Big( \sum_{k=1}^n D_{kk}\boldsymbol{e}_k \boldsymbol{e}_k^T \Big) B^T \boldsymbol{c}_j + (\eta_j - \delta_j)^2 \\
&= \lambda_{min}(D)\boldsymbol{c}_i^T BDB^T \boldsymbol{c}_i + (\eta_i - \delta_i)^2 \\
&\quad + \lambda_{min}(D)\boldsymbol{c}_j^T BDB^T \boldsymbol{c}_j + (\eta_j - \delta_j)^2 \\
&= \lambda_{min}(D)\boldsymbol{c}_i^T S \boldsymbol{c}_i + (\eta_i - \delta_i)^2 + \lambda_{min}(D)\boldsymbol{c}_j^T S \boldsymbol{c}_j + (\eta_j - \delta_j)^2 \\
&= \lambda_{min}(D)\eta_i + (\eta_i - \delta_i)^2 + \lambda_{min}(D)\eta_j + (\eta_j - \delta_j)^2 \\
&\geq \lambda_{min}(D)\frac{(\eta_i + \eta_j)}{4} + \frac{|\delta_i|}{4} \min\Big( |\delta_i|, \lambda_{min}(D) \Big) \\
&\quad + \frac{|\delta_j|}{4} \min\Big( |\delta_j|, \lambda_{min}(D) \Big)
\end{aligned}
$$

$$(19)$$

after substituting $\eta_i = \boldsymbol{c}_i^T S \boldsymbol{c}_i \geq \lambda_{min}(S)\|\boldsymbol{c}_i\|^2$, $\eta_j = \boldsymbol{c}_j^T S \boldsymbol{c}_j \geq \lambda_{min}(S)\|\boldsymbol{c}_j\|^2$ and using Lemma A.4 that gives us a lower bound for the second and third term. Thus, we obtain:

$$
\begin{aligned}
\|S_i - S_j\|_F^2 &\geq \frac{\lambda_{min}(D)\Big( \lambda_{min}(S)(\|\boldsymbol{c}_i\|^2 + \|\boldsymbol{c}_j\|^2) \Big)}{4\|B^{-1}\|_F^4} + \frac{|\delta_i|}{4\|B^{-1}\|_F^4} \min\Big( |\delta_i|, \lambda_{min}(D) \Big) \\
&\quad + \frac{|\delta_j|}{4\|B^{-1}\|_F^4} \min\Big( |\delta_j|, \lambda_{min}(D) \Big) \\
&\geq \frac{\lambda_{min}^2(D)\Big( \|\boldsymbol{c}_i\|^2 + \|\boldsymbol{c}_j\|^2 \Big)}{4\|B^{-1}\|_F^4} + \frac{|\delta_i|}{4\|B^{-1}\|_F^4} \min\Big( |\delta_i|, \lambda_{min}(D) \Big) \\
&\quad + \frac{|\delta_j|}{4\|B^{-1}\|_F^4} \min\Big( |\delta_j|, \lambda_{min}(D) \Big),
\end{aligned}
$$

$$(20)$$

where $\lambda_{min}(S) \geq \lambda_{min}(D)\lambda_{min}^2(B) = \lambda_{min}(D)$ ($\because \lambda_{min}(ST) \geq \lambda_{min}(S)\lambda_{min}(T)$ and $\lambda_{min}(B) = \lambda_{min}(B^T) = 1$ since it is a lower triangular matrix where all diagonal entries are 1) and $\|ST\|_F \leq \|S\|_F\|T\|_F$. For the case when one of the covariance matrices is $S$ corresponding to no intervention and the other is $S_i$ corresponding to intervention on node $V_i$, then from Eq. 14 and

the following similar analysis we have:

$$\|B^{-1}(S_i - S)B^{-T}\|_F^2 \geq \left[\sum_{k=1}^{n}(D_{kk}\boldsymbol{c}_i^T B\boldsymbol{e}_k)^2 + (\eta_i - \delta_i)^2\right]$$

$$\geq \lambda_{min}(D)\frac{\eta_i}{4} + \frac{|\delta_i|}{4}\min\left(|\delta_i|, \lambda_{min}(D)\right) \qquad (21)$$

$$\|S_i - S\|_F^2 \geq \frac{\lambda_{min}^2(D)\|\boldsymbol{c}_i\|^2 + |\delta_i|\min\left(|\delta_i|, \lambda_{min}(D)\right)}{4\|B^{-1}\|_F^4},$$

thereby completing our proof of this lemma.

$\square$

**Lemma A.4.** *Given $\lambda \geq 0$ and $\eta \geq 0$, the following inequality holds true:*

$$\lambda\eta + (\eta - \delta)^2 \geq \frac{\lambda\eta}{4} + \frac{|\delta|}{4}\min\left(|\delta|, \lambda\right). \qquad (22)$$

*Proof. Case 1:* $\left(\delta < 0\right)$ Let $T \triangleq \lambda\eta + (\eta - \delta)^2$, then we have $T \geq \lambda\eta + \delta^2 \geq \lambda\eta + \delta^2/4$.

*Case 2:* $\left(\delta > 0 \text{ and } [0 \leq \eta < \delta/2 \text{ or } \eta > 3\delta/2]\right)$ Again in this case $T \geq \lambda\eta + \delta^2/4$.

*Case 3:* $\left(\delta > 0 \text{ and } \delta/2 \leq \eta \leq 3\delta/2\right)$ In this case, we have $T \geq \lambda\eta$. Also, we have $\eta > \delta/2 \implies T \geq \lambda\delta/2$ which together implies:

$$T \geq \max\left(\lambda\eta, \lambda\frac{\delta}{2}\right)$$
$$\geq \frac{\lambda\eta}{2} + \frac{\lambda\delta}{4}. \qquad (23)$$

Thus, combining the above three cases we have the following lower bound on the value of $T$:

$$T \geq \min\left(\lambda\eta + \frac{\delta^2}{4}, \frac{\lambda\eta}{2} + \frac{|\delta|}{4}\right)$$
$$\geq \frac{\lambda\eta}{4} + \min\left(\frac{\delta^2}{4}, \frac{\lambda|\delta|}{4}\right) \qquad (24)$$
$$= \frac{\lambda\eta}{4} + \frac{|\delta|}{4}\min\left(|\delta|, \lambda\right),$$

which completes the proof. $\square$

### A.4   Proof of Lemma A.3

*Proof.* From Eq. 3, we have $S_i = B_i D_i B_i^T$ where $B_i = (I - A + \boldsymbol{e}_i\boldsymbol{c}_i^T)^{-1}$ and $D_i = D - (\sigma_i - \sigma_i')\boldsymbol{e}_i\boldsymbol{e}_i^T \triangleq D - \delta_i\boldsymbol{e}_i\boldsymbol{e}_i^T$. Since $\boldsymbol{e}_i\boldsymbol{c}_i^T$ is a rank-1 update to the (I-A) matrix, using the Sherman-Morrison identity we obtain:

$$B_i = (I - A + \boldsymbol{e}_i\boldsymbol{c}_i^T)^{-1}$$
$$= (I - A)^{-1} - \frac{(I - A)^{-1}\boldsymbol{e}_i\boldsymbol{c}_i^T(I - A)^{-1}}{1 + \boldsymbol{c}_i^T(I - A)^{-1}\boldsymbol{e}_i}$$
$$= B - \frac{B\boldsymbol{e}_i\boldsymbol{c}_i^T B}{1 + \boldsymbol{c}_i^T B\boldsymbol{e}_i} \qquad (25)$$
$$= B - B\boldsymbol{e}_i\boldsymbol{c}_i^T B$$
$$\triangleq B - \boldsymbol{r}_i\boldsymbol{q}_i^T,$$

where $\boldsymbol{r}_i \triangleq \frac{B\boldsymbol{e}_i}{d_i}$, scalar $d_i \triangleq 1 + \boldsymbol{c}_i^T B\boldsymbol{e}_i = 1$ and $\boldsymbol{q}_i \triangleq B^T\boldsymbol{c}_i$. The scalar $d_i = 1$ since $\boldsymbol{q}_i^T\boldsymbol{e}_i = \boldsymbol{c}_i^T B\boldsymbol{e}_i = 0$ given by the following lemma:

**Lemma A.5.** *Let A and B be a lower triangular matrix and $c_i$ be a vector such that $c_i = a_i - a_i'$ and $[c_i]_t = 0, \forall t \geq i$ (see Atomic Interventions paragraph of §3 for definition), then $q_i^T e_i = c_i^T B e_i = 0$. Also, if $c_i \neq 0$, then $q_i = B^T c_i \neq 0$.*

The proof of the above lemma can be found below after the proof of the current lemma. Next, $S_i$ is given by:

$$\begin{aligned}
S_i &= B_i D_i B_i^T \\
&= B_i(D - \delta_i e_i e_i^T)B_i^T \\
&= \underbrace{B_i D B_i^T}_{\text{term 1}} - \delta_i \underbrace{B_i e_i e_i^T B_i^T}_{\text{term 2}}.
\end{aligned} \tag{26}$$

If the intervened node $V_i$ is one of *root* nodes of the underlying unknown causal graph, then $c_i = \mathbf{0} \implies q_i = B^T c_i = \mathbf{0} \implies B_i = B$ (from Eq. 25), and thus:

$$B_i D B_i^T = B D B^T = S, \tag{27}$$

where $S$ is the covariance matrix of the observational distribution. Otherwise, for non-root nodes, *term 1* in the above equation is:

$$\begin{aligned}
B_i D B_i &= (B - r_i q_i^T)D(B - r_i q_i^T)^T \\
&= B D B^T - B D q_i r_i^T - (B D q_i r_i^T)^T + r_i q_i^T D q_i r_i^T \\
&= S - w_i r_i^T - r_i w_i^T + \eta_i r_i r_i^T
\end{aligned} \tag{28}$$

where $w_i = B D q_i$ and scalar $\eta_i = q_i^T D q_i$. Next simplifying *term 2* in Eq. 26 we get:

$$\begin{aligned}
B_i e_i (B_i e_i)^T &= \left(B e_i - r_i q_i^T e_i\right)\left(B e_i - r_i q_i^T e_i\right)^T \\
&= B e_i e_i^T B^T = r_i r_i^T
\end{aligned} \tag{29}$$

since $q_i^T e_i = c_i^T B e_i = 0$ from Lemma A.5 and $r_i \triangleq B e_i$ (from Eq. 25). Thus the covariance matrix of the endogenous variables in the intervened distribution is given by:

$$S_i = \begin{cases} S - \delta_i r_i r_i^T, & \text{for root node} \\ B_i D B_i^T - \delta_i r_i r_i^T, & \text{otherwise} \end{cases} \tag{30}$$

thereby completing our proof. $\qquad\square$

*Proof of Lemma A.5.* Showing $q_i \neq \mathbf{0}$: We are given that $B$ is a lower triangular matrix with non-zero diagonal entries. Also, $c_i = a_i - a_i' \neq 0$ and $[c_i]_t = 0, \forall t \geq i \implies \exists t \text{ s.t. } [c_i]_t \neq 0$ and let $t^*$ be that last index where $c_i$ is non-zero. Now lets look at the $[B^T c_i]_{t^*} = c_{t^*} \cdot [B^T]_{t^*,t^*} \neq 0$ since $B^T$ is a upper triangular matrix and $[c_i]_{t^*} \neq 0$.

Showing $q_i^T e_i = 0$: $B e_i$ is the $i^{th}$ column of the lower triangular matrix $B \implies [B e_i]_t = 0, \forall t < i \implies c_i^T B e_i = 0$ since $[c_i]_t = 0, \forall t \geq i$. $\qquad\square$

## A.5 Proof of Lemma A.2

*Proof.* Similar to the proof of Lemma A.1, without loss of generality, we will assume that the endogenous variables are topologically ordered based on the underlying causal graph such that the adjacency matrix $A$ is lower triangular. The permutation matrix will only permute the rows of the mean vectors $m_i$ and $m_j$; hence, there will be no effect on the value of $\|m_i - m_j\|_F^2$.

Now, let the mean of the Gaussian distribution corresponding to intervention on node $V_i$ be given by (see Atomic Intervention paragraph in §3 for definition):

$$\begin{aligned}
m_i = B_i \mu_i &= (B - B e_i c_i^T B)(\mu + \gamma_i e_i) \\
&\implies B^{-1} m_i = (I - e_i c_i^T B)(\mu + \gamma_i e_i),
\end{aligned} \tag{31}$$

where $B_i = (I - A_i)^{-1} = B - B e_i c_i^T B$ (from Lemma A.3), $\mu_i$ is the new mean vector for the noise distribution, $\mu$ is the mean vector of the observational noise distribution and $\gamma_i e_i$ is the update

to the mean of the noise distribution when intervening on node $V_i$. Similarly, the mean corresponding to the intervened distribution on node $V_j$ is given by:

$$
\begin{aligned}
\boldsymbol{m}_j = B_j \boldsymbol{\mu}_j &= (B - B\boldsymbol{e}_j \boldsymbol{c}_j^T B)(\boldsymbol{\mu} + \gamma_j \boldsymbol{e}_j) \\
\implies B^{-1} \boldsymbol{m}_j &= (I - \boldsymbol{e}_j \boldsymbol{c}_j^T B)(\boldsymbol{\mu} + \gamma_j \boldsymbol{e}_j).
\end{aligned}
\tag{32}
$$

Looking at the $i^{\text{th}}$ entry of the vector $B^{-1}\boldsymbol{m}_i$, we get:

$$
\boldsymbol{e}_i^T B^{-1} \boldsymbol{m}_i = \boldsymbol{e}_i^T \boldsymbol{\mu} - \boldsymbol{c}_i^T B \boldsymbol{\mu} + \gamma_i - \cancelto{0}{\boldsymbol{c}_i^T B \boldsymbol{e}_i},
\tag{33}
$$

and the $i^{\text{th}}$ entry of the vector $B^{-1}\boldsymbol{m}_j$ is given by:

$$
\boldsymbol{e}_i^T B^{-1} \boldsymbol{m}_j = \boldsymbol{e}_i^T \boldsymbol{\mu}.
\tag{34}
$$

Similarly, the $j^{\text{th}}$ entry of the vector $B^{-1}\boldsymbol{m}_i$ is given by:

$$
\boldsymbol{e}_j^T B^{-1} \boldsymbol{m}_i = \boldsymbol{e}_j^T \boldsymbol{\mu},
\tag{35}
$$

and the $i^{\text{th}}$ entry of the vector $B^{-1}\boldsymbol{m}_j$ is given by:

$$
\boldsymbol{e}_j^T B^{-1} \boldsymbol{m}_j = \boldsymbol{e}_j^T \boldsymbol{\mu} - \boldsymbol{c}_j^T B \boldsymbol{\mu} + \gamma_j - \cancelto{0}{\boldsymbol{c}_j^T B \boldsymbol{e}_j}.
\tag{36}
$$

Thus, the difference in the mean of both distributions can be lower bounded by:

$$
\|B^{-1}(\boldsymbol{m}_i - \boldsymbol{m}_j)\|_F^2 \geq (\gamma_i - \boldsymbol{c}_i^T B \boldsymbol{\mu})^2 + (\gamma_j - \boldsymbol{c}_j^T B \boldsymbol{\mu})^2.
\tag{37}
$$

Now based on the value of $|\gamma|_i$ and $|\boldsymbol{c}_i^T B \boldsymbol{\mu}|$, we have:

$$
(\gamma_i - \boldsymbol{c}_i^T B \boldsymbol{\mu})^2 \geq
\begin{cases}
\frac{\gamma_i^2}{4}, & \psi_i^+ \triangleq |\boldsymbol{c}_i^T B \boldsymbol{\mu}| < \frac{|\gamma_i|}{2} \text{ or } |\boldsymbol{c}_i^T B \boldsymbol{\mu}| > \frac{3|\gamma_i|}{2} \\
0, & \psi_i^- \triangleq \frac{|\gamma_i|}{2} \leq |\boldsymbol{c}_i^T B \boldsymbol{\mu}| \leq \frac{3|\gamma_i|}{2},
\end{cases}
\tag{38}
$$

where $\psi_i^+$ and $\psi_i^-$ are used to denote two different mutually exclusive different events for ease of exposition later. In the case when $\psi_i^-$ is true, we have:

$$
\begin{aligned}
\frac{\gamma_i^2}{4} &\leq (\boldsymbol{c}_i^T B \boldsymbol{\mu})^2 \leq \|\boldsymbol{c}_i\|^2 \|B \boldsymbol{\mu}\|^2 \qquad \text{(Cauchy-Schwarz)} \\
\frac{\gamma_i^2}{4\|B\boldsymbol{\mu}\|^2} &\leq \|\boldsymbol{c}_i\|^2.
\end{aligned}
\tag{39}
$$

Similarly, we have:

$$
(\gamma_j - \boldsymbol{c}_j^T B \boldsymbol{\mu})^2 \geq
\begin{cases}
\frac{\gamma_j^2}{4}, & \psi_j^+ \triangleq |\boldsymbol{c}_j^T B \boldsymbol{\mu}| < \frac{|\gamma_j|}{2} \text{ or } |\boldsymbol{c}_j^T B \boldsymbol{\mu}| > \frac{3|\gamma_j|}{2} \\
0, & \psi_j^- \triangleq \frac{|\gamma_j|}{2} \leq |\boldsymbol{c}_j^T B \boldsymbol{\mu}| \leq \frac{3|\gamma_j|}{2},
\end{cases}
\tag{40}
$$

and in the event when $\psi_j^-$ is true, we have:

$$
\begin{aligned}
\frac{\gamma_j^2}{4} &\leq (\boldsymbol{c}_j^T B \boldsymbol{\mu})^2 \leq \|\boldsymbol{c}_j\|^2 \|B \boldsymbol{\mu}\|^2 \qquad \text{(Cauchy-Schwarz)} \\
\frac{\gamma_j^2}{4\|B\boldsymbol{\mu}\|^2} &\leq \|\boldsymbol{c}_j\|^2.
\end{aligned}
\tag{41}
$$

Combining Eq. 38 and 40 and using $\|B^{-1}(\boldsymbol{m}_i - \boldsymbol{m}_j)\|_F \leq \|B^{-1}\|_F \|\boldsymbol{m}_i - \boldsymbol{m}_j\|_F$, we have:

$$
\|\boldsymbol{m}_i - \boldsymbol{m}_j\|_F^2 \geq
\begin{cases}
\frac{\gamma_i^2}{4\|B^{-1}\|_F^2} + \frac{\gamma_j^2}{4\|B^{-1}\|_F^2}, & \psi_i^+, \psi_j^+ \text{ are active} \\
\frac{\gamma_i^2}{4\|B^{-1}\|_F^2}, & \psi_i^+, \psi_j^- \text{ are active, Eq. 41 holds} \\
\frac{\gamma_j^2}{4\|B^{-1}\|_F^2}, & \psi_i^-, \psi_j^+ \text{ are active, Eq. 39 holds} \\
0, & \psi_i^-, \psi_j^- \text{ are active, Eq. 39 and 41 holds}.
\end{cases}
\tag{42}
$$

For the case when one of the mean say $\boldsymbol{m}_j$ is observational then:

$$\boldsymbol{m}_j = \boldsymbol{m} = B\boldsymbol{\mu}$$
$$\implies B^{-1}\boldsymbol{m}_j = \boldsymbol{\mu}. \tag{43}$$

Thus the $i^{\text{th}}$ and $j^{\text{th}}$ entry of $\boldsymbol{m}_j$ is $\boldsymbol{e}_i^T\boldsymbol{\mu}$ and $\boldsymbol{e}_j^T\boldsymbol{\mu}$, respectively. Thus using Eq. 33 and 35 we get:

$$\|B^{-1}(\boldsymbol{m}_i - \boldsymbol{m}_j)\|_F^2 \geq (\gamma_i - \boldsymbol{c}_i^T B\boldsymbol{\mu})^2. \tag{44}$$

Then again, following a similar analysis as in Eq. 38 and 39 we have the following cases:

$$\|\boldsymbol{m}_i - \boldsymbol{m}_j\|_F^2 \geq \begin{cases} \frac{\gamma_i^2}{4\|B^{-1}\|_F^2}, & \psi_i^+ \text{is active} \\ 0, & \psi_i^- \text{ is active, Eq. 39 holds,} \end{cases} \tag{45}$$

which is equivalent to say that only case 2 and 4 are applicable in Eq. 42. This completes the proof. $\qquad\square$

# B  Setup and Additional Empirical Result

## B.1  Experimental Setup Discussion - Simulation

**Random Graph Generation:**  As mentioned in the Simulation setup in §6, we randomly generate the adjacency matrix $A$ of the causal graphs used to simulate the mixture distribution. All the weights in the adjacency matrix are sampled in the range $[-1, -0.5] \cup [0.5, 1]$ bounded away from zero. This gives us a complete graph of all nodes. Thus, to introduce sparsity in the graph, we only keep an edge with probability $\zeta$ by setting the corresponding value in the adjacency matrix to zero if the edge is removed. Unless otherwise specified, we set $\zeta = 0.8$ for all our experiments.

**Step 1 of Alg. 1:**  We use the GaussianMixture class from the scikit-learn Python package to disentangle the components of the mixture. For all our experiments, we use the default $tol = 10^{-3}$ used by GaussianMixture to decide on convergence of the underlying EM algorithm.

**Step 2 of Alg. 1:**  We run the UT-IGSP algorithm in Step 2 of our Alg. 1 with the standard parameter mentioned in the documentation. Specifically, we use $\alpha = 10^{-3}$ for both *MemoizedCITester* and *MemoizedInvarianceTester* functions used by UT-IGSP.

**Specific to results in Appendix**  Unless otherwise specified, for all the results in the appendix we run all the experiments for 5 random settings. We ignore the error bars in the appendix for clarity in exposition. Also for all the experiments, the half-intervention setting i.e. where the mixture contains intervention on half of the randomly selected nodes is considered.

## B.2  Experimental Setup Discussion - Biology Dataset

**Dataset:**  In the interventional data the signaling protein is also perturbed along with the receptor enzymes. The different perturbed signaling proteins (along with the number of samples corresponding) are: Akt (911), PKC (723), PIP2 (810), Mek (799), PIP3 (848). The observational data contained 1755 samples so overall 5846 samples were used for our experiments. For details see Wang et al. [31] and Sachs et al. [22].

## B.3  Code

The source code to all the experiments can be found in the following GitHub repository: https://github.com/BigBang0072/mixture_mec

## B.4  Computational Resources

We use an internal cluster of CPUs to run all our experiments. We run 10 random runs for each of the experimental configurations and report the mean (as points) and $5^{th}$ and $95^{th}$ quantiles as error bars for all our experiments in the main paper, and we report only mean the mean for the experiments in the appendix to declutter the figures.

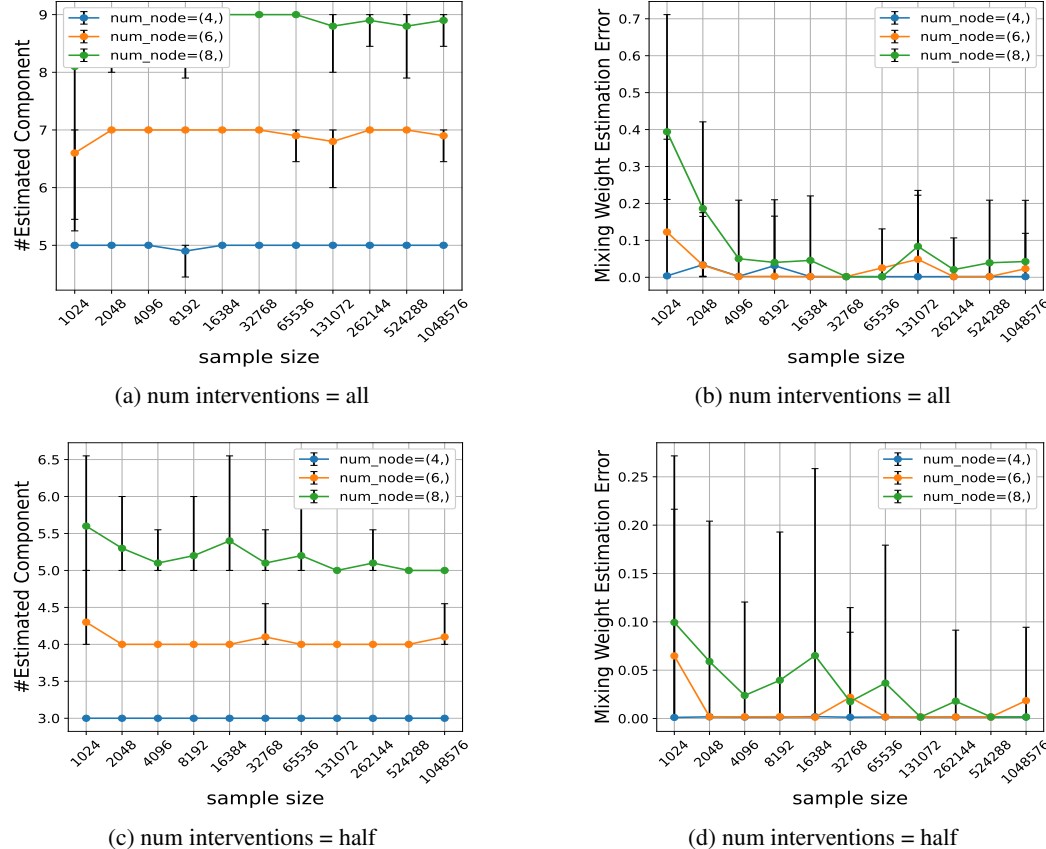

(a) num interventions = all

(b) num interventions = all

(c) num interventions = half

(d) num interventions = half

Figure 2: **Other Evaluations Metrics for the simulation experiments in Fig. 1**: The top row denotes the corresponding metrics for *all* interventions in the mixture setting and the bottom row to the *half* setting. The first column shows the number of components estimated by our algorithm Mixture-UTIGSP. For the *all* setting, the actual number of components corresponding to the system with nodes 4,6 and 8 are 5,7,9 respectively (one intervention on each node + one observational distribution). We observe that Mixture-UTIGSP is able to correctly estimate the number of components even with a small number of samples. Similarly, for *half* setting, the actual number of components corresponding to the system with nodes 4,6 and 8 are 3,4,5 respectively (intervention on half of node and one observational distribution). Even for this case Mixture-UTIGSP is able to correctly estimate the number of components. The second column shows the error in the estimation of the mixing coefficient ($\pi_i$'s, see Definition 4.1). For both cases, we observe that the error in the estimation of the mixing coefficient goes to zero as the sample size increases.

## B.5 Evaluation Metric

**Parameter Estimation Error** Given the mixture distribution, the first step of our Alg. 1 estimates the parameters of every interventional and observational distribution present in the mixture (mixing weight $\pi_i$, mean $\boldsymbol{m}_i$ and covariance matrix $S_i$). Alg. 1 returns the maximum number of possible components, i.e. $k = n + 1$, by default. For all our experiments, we used this default value and left searching over the number of components for future work. To calculate the parameter estimation error, we first find the best match of the estimated parameters with the ground truth parameters based on the minimum error between the mean and covariance matrix. More specifically, let $k^*$ be the ground truth number of components in the mixture. Then, we iterate over all possible $k^*$ sized subsets of the estimated parameters and choose the one with the smallest absolute error sum between the mean and covariance matrix. Formally:

$$\text{Parameter Estimation Error} \triangleq \min \left\{ \sum_{i=1}^{k^*} \left( |\boldsymbol{m}_i - \hat{\boldsymbol{m}}_{\rho(i)}| + |S_i - \hat{S}_{\rho(i)}| \right) : \rho \in perm([n+1]) \right\} \quad (46)$$

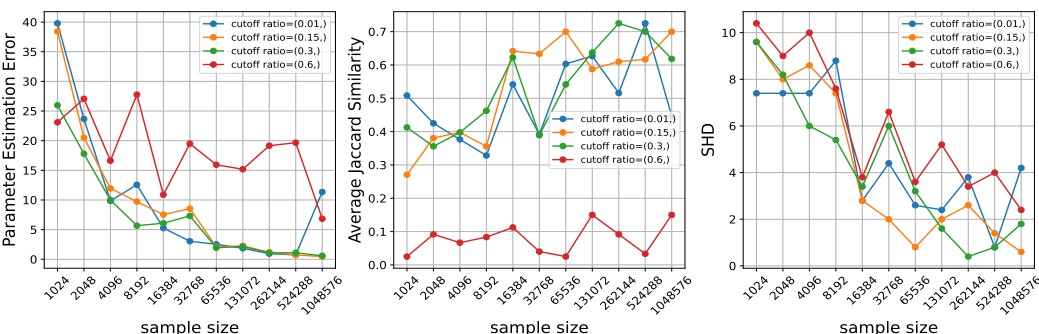

Figure 3: **Performance of Alg. 1 as we change the cutoff ratio used for automatic component selection**: We consider graphs with 6 nodes in this experiment with *half* intervention setting. In step 2 of Mixture-UTIGSP, we select the number of components using the log-likelihood curve. We scan the curve starting from the mixture model with the largest number of components to the smallest and stop where the relative change in the likelihood increases above a cutoff ratio (to select the elbow point of the curve). The cutoff ratio in the algorithm is chosen to be an arbitrary number close to zero. Here we compare the performance of Mixture-UTIGSP on all three metrics for the half setting of Fig. 1 as we vary the cutoff ratio. We observe that for the cutoff ratio close to zero i.e. 0.01, 0.15,0.3 the performance remains similar showing that the model selection criteria are robust to the selected cutoff ratio. The number of nodes

where $perm([n+1])$ represents all possible permutations of the indices $[0, 1, \ldots, n]$ and $\rho^*$ is the corresponding permutation with the minimum error.

**Average Jaccard Similarity (JS)**   In step 2 of our Alg. 1, UT-IGSP estimates the unknown interventional targets for each of the individual components disentangled in Step 1. We use the same matching ($\rho^*$) found in the parameter estimation error step (as mentioned above) to calculate the Jaccard similarity between the estimated and ground truth intervention target for that component. Formally:

$$\text{Avg. Jaccard Similarity} \triangleq \frac{1}{k^*} \sum_{i=1}^{k^*} JS(\boldsymbol{t}_i, \hat{\boldsymbol{t}}_{\rho(i)}) = \frac{1}{k^*} \sum_{i=1}^{k^*} \frac{|\boldsymbol{t}_i \cap \hat{\boldsymbol{t}}_{\rho^*(i)}|}{|\boldsymbol{t}_i \cup \hat{\boldsymbol{t}}_{\rho^*(i)}|}, \tag{47}$$

where $\boldsymbol{t}_i$ is the ground truth intervention target set and $\hat{\boldsymbol{t}}_j$ is the estimated target set. For the case when both $\boldsymbol{t}_i = \hat{\boldsymbol{t}}_i = \phi$, then $JS(\boldsymbol{t}_i, \hat{\boldsymbol{t}}_i) \triangleq 1$.

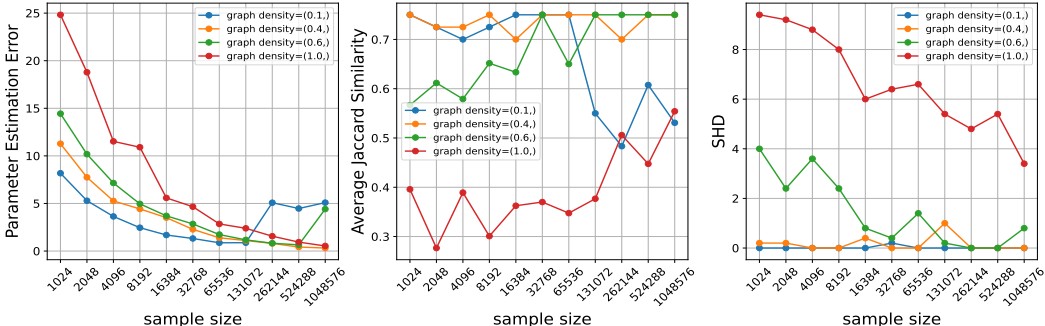

Figure 4: **Performance of Alg. 1 as we change the density of the underlying true causal graph**: The mixture data contains atomic interventions on all nodes as well as observational data (*half* setting as described in the results in §6). The column shows different evaluation metrics, i.e., Parameter Estimation Error, Average Jaccard Similarity, and SHD (see Evaluation metric paragraph in §6). In this experiment, we vary the density of the underlying causal graph by keeping the edges in a fully connected graph with a fixed probability, labeled as density in the legend of the above plots (see random graph generation paragraph in §B.1 for details). The maximum possible density is 1, i.e., the probability of keeping an edge is 1, corresponding to a fully connected graph, and the lowest possible density is 0. We observe that as the density of the graph increases, we require more samples to achieve similar performance to less dense graphs on all three metrics. Our Theorem 4.1 shows that the sample complexity required for estimating the parameters of the mixture is proportional to the norm of the adjacency matrix $\|A\|$ and as the density of the graph increases $\|A\|$ increases. Thus, as the density increases, we require more samples to achieve a similar performance in estimating the parameters of the mixture, as seen in the parameter estimation error plot above.

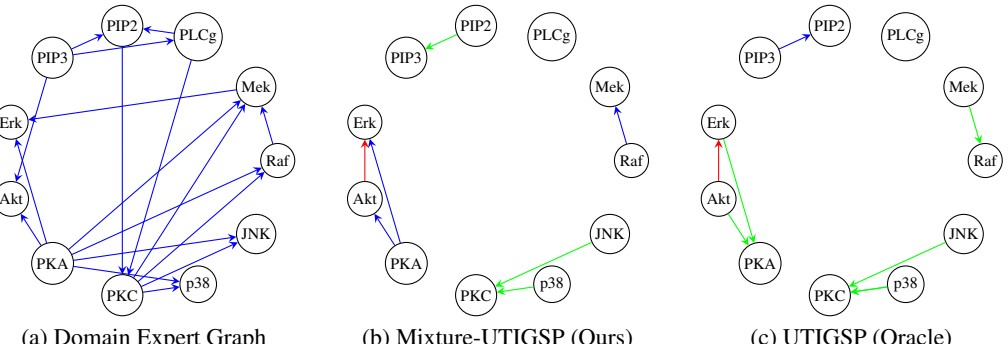

(a) Domain Expert Graph      (b) Mixture-UTIGSP (Ours)      (c) UTIGSP (Oracle)

Figure 5: **Ground truth and estimated causal graph for Protein signaling dataset [22]**: Fig 5a is the graph created with the help of domain experts for this problem [31]. 5b shows the graph estimated by our Mixture-UTIGSP and 5c is the graph estimated by oracle UT-IGSP when they are given the ground truth disentangled mixture. The blue colored arrow in 1b and 1c shows the correctly recovered edges in the domain expert graph. Green shows the edges with the same skeleton in the domain expert graph but in a reversed direction. The red shows the edges that are incorrectly added in the estimated graph. We observe that Mixture-UTIGSP correctly identifies two more edges (PKA->ERK and PKA-> Akt) as compared to an oracle which could be due to randomness in the UTIGSP algorithm. For this estimation, the best-performing cutoff of 0.01 was selected (see Table 1).

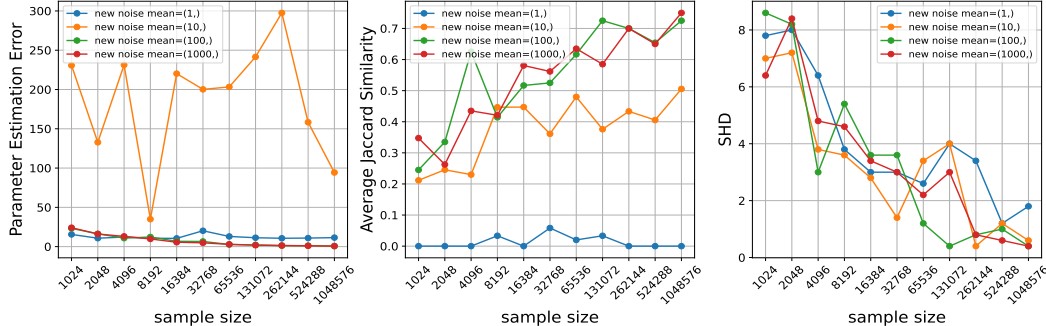

(a) Varying the mean of the noise distribution ($|\gamma_i|$): The initial mean of the noise distribution is 0.0 for all the nodes. Upon intervening on a node to generate the interventional distribution, we change the mean of the noise to a different value. The variance of the noise distribution of the intervened node is kept the same as the initial distribution i.e. 1.0. From Theorem 4.1, we expect that as the new mean increases further from the initial mean = 0.0, the parameter estimation error should be lower for a given sample size and lead to better performance in intervention target estimation and causal discovery. As expected, the setting with the smallest change in the mean of the noise distribution (blue curve) has the worst performance. The case when the new noise mean is 10.0 (orange curve) is unusual where we see an unexpected increase in parameter estimation error.

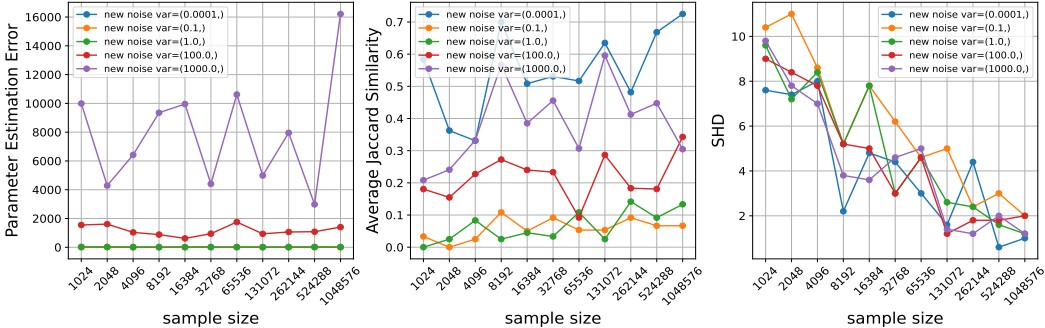

(b) Varying the variance of the noise distribution ($|\delta_i|$): The initial variance of the noise distribution is 1.0 for all nodes, and we change the variance of the new noise distribution upon intervention in this experiment. The mean of the noise distribution of the intervened distribution is kept the same as the initial distribution i.e. 0.0. From Theorem 4.1, we see that sample complexity to recover the parameters of the mixture distribution is inversely proportional to the change in the noise variance $|\delta_i|$. Thus, we expect that the performance of Alg. 1 should improve as the new noise variance moves away from the initial noise variance 1.0. We can see in the above plot that the performance of the green curve ($\delta_i = 0$) is worst in terms of the Jaccard similarity and SHD of the recovered graph, validating our expectation. Parameter estimation cannot be directly compared since as the variance increases, the norm of the covariance matrix increases, and thus, the overall error in the estimation error increases. We observe that compared to changing the mean (Fig. 6a) increasing the variance gives slightly lower performance gains.

Figure 6: **Performance of Alg. 1 as we change different parameters of interventions**: We consider graphs with 6 nodes in this experiment. The mean of all noise distributions without any intervention is 0.0, and the variance is 1.0. The mixture data contains atomic interventions on all nodes and observational data (*half* setting as described in results in §6). The column shows different evaluation metrics, i.e., Parameter Estimation Error, Average Jaccard Similarity, and SHD (see evaluation metric paragraph in §6). From Theorem 4.1, we observe that the sample complexity for recovering the parameters of the mixture is inversely proportional to the change in the mean of the noise distribution $\gamma_i^2$ and change in the variance of the noise distribution $|\delta_i|$. In this experiment, we vary these two parameters one at a time and empirically validate this observation.

