# OpenReview forum: "Learning Mixtures of Unknown Causal Interventions"
_NeurIPS.cc/2024/Conference — NeurIPS 2024 poster_

### Official Review · Reviewer_SSWb · 2024-06-20

**Soundness:** 4
**Presentation:** 3
**Contribution:** 3
**Rating:** 8
**Confidence:** 5

**Summary:**

This paper explores the challenge of disentangling and identifying causal relationships in situations where interventional data is noisy and mixed with both intended and unintended effects. It focuses on applying interventions within linear SEMs with Gaussian noise without prior knowledge of the true causal graph.  It presents an efficient algorithm that can learn and separate individual components of a mixture of unknown interventions, allowing for the recovery of true interventional distributions even when intervention targets are unknown.

**Strengths:**

This paper addresses a relatively unexplored problem of disentangling mixed interventional and observational data within linear SEMs without prior knowledge of the causal graph. This is a challenging problem with significant implications for causal inference, making the study innovative and important for the field.
The authors propose an efficient algorithm that leverages the properties of linear SEMs with Gaussian noise to recover individual components of the mixture. The methodological approach is rigorous, with detailed theoretical support including proofs of identifiability and sample complexity. This thorough theoretical treatment provides a solid foundation for the claims made in the paper.
The paper is well-structured with clear explanations of the problem, methodology, and results.

**Weaknesses:**

The methods developed in the paper are specific to linear SEMs with Gaussian noise, which might limit their applicability in scenarios where these assumptions do not hold. Non-linear relationships or non-Gaussian noise structures, which are common in many real-world datasets, may not be adequately addressed by the proposed approach.

**Questions:**

1. I suggest improving readability by properly describing insights related to theorems, even though some of the proofs focus on matrix computations.
2. In the experimental setup, what would be the effect of different variances in Gaussian noise?
3.There might be some symbols used incorrectly:
In line 104, "the edge U" should be checked.
In line 128, the definition of M_I needs verification.
In line 134, the letter \(\kappa\) is used when mentioning shift interventions, but \(\gamma\) is used below. I know that do-interventions also change \(\gamma\), but I suggest keeping the notation consistent.
In line 279, "A_{ij} > 0" should possibly be "|A_{ij}| > 0".

**Limitations:**

The author has explained the limitations of the study. This paper has no possible negative social impact.

---

> ### Author Rebuttal · Authors · 2024-08-07
>
> We thank the reviewer for their time and thoughtful comments. Below, we answer the weaknesses and questions raised by the reviewer.
>
> >Weakness
>
> **Weakness 1: Experiments**
>
> *Experiment on the real-world dataset:* We have run our algorithm on a real-world protein signaling network dataset. Our method performs equivalently on all the metrics compared to the Oracle algorithm, which has access to the disentangled data (please refer to the **Evaluation on real-world dataset** paragraph in the Global comment for details). Also, In the **additional pdf document** attached to the global comment, we provide the ground truth graph and the estimated graph from both algorithms.
>
> *Experiment with Linear SEM with non-gaussian noise*:
> The main motivation for our work comes from the problem of causal discovery. In general, one can only identify the causal graph up to their Markov Equivalence Class using only observational data. Thus, we need interventional data to fully identify the causal graph. The intervention targets are sometimes noisy, so we can only obtain a mixture of interventional data.
> However, Shimizu et al. [1] showed that observational data is **sufficient for learning the underlying causal graph when the data-generating process is a linear SEM with additive non-Gaussian noise** with no latent confounders.  They also proposed an algorithm (LINGAM)  that uses Non-Linear ICA over observational data to identify the causal graph. Thus, Linear SEM with non-gaussian noise doesn't require interventional data to identify the underlying causal graph. Hence, studying the mixture of interventional distribution for this framework is an interesting problem outside the scope of current work.
>
>
> \
> \
> >Questions
>
> **Questions 1**: We thank the reviewer for this comment. We will add the intuition behind the proofs in the final version of the paper.
> \
> \
> **Question 2: (Effect of different variance in Gaussian noise)**
>
> We have an ablation experiment in Fig 5b of Appendix B.4, where we study the effect of changing the variance of the noise distribution post-intervention. The initial variance of the noise distribution of every node is 1.0, and post-intervention, we set the variance of the intervened node to take values from the set $\{0.1,1.0,4.0,8.0\}$. If the final noise variance (after intervention) is close to the initial noise variance (1.0), then the Jaccard similarity and SHD of the estimated targets and causal graphs are worse. This is also expected from our theoretical result (Theorem 4.1), which states that the sample complexity to recover the parameters of the mixture distribution is inversely proportional to the change in the noise variance ($\delta_i$ = |final variance - initial variance|).
> \
> \
> **Typos**: We thank the reviewer for pointing out the typos and other writing suggestions in the text. We will incorporate them into the final version of the paper.
>
>
> \
> \
> [1] Shohei Shimizu, Patrik O. Hoyer, Aapo Hyvärinen, and Antti Kerminen. A linear non-gaussian acyclic model for causal discovery. Journal of Machine Learning Research, 7(72):2003–2030, 2006

---

> > ### Comment · Reviewer_SSWb · 2024-08-12
> >
> > Thanks for your response, it addresses my concerns. I will keep my rating.

---

### Official Review · Reviewer_nhPx · 2024-07-13

**Soundness:** 3
**Presentation:** 3
**Contribution:** 3
**Rating:** 6
**Confidence:** 5

**Summary:**

The paper proposes a method to disentangle the observational and interventional data under linear SEM with Gaussian noise.

**Strengths:**

The algorithm proposed in this paper efficiently disentangles components of mixtures arising from unknown interventions, accommodating both soft and hard interventions. The proposed method is supported by thorough theoretical analysis. Additionally, the paper is well-written and easy to follow.

**Weaknesses:**

The experiments are not comprehensive enough to thoroughly assess the effectiveness of the proposed method. More experiments on real-world datasets are required. This is especially necessary since the problem setup for the mixture of interventions is a bit restrictive. Also, it would be interesting to analyze how the method empirically performs on datasets that are not generated from Linear SEM or do not have Gaussian noises.

**Questions:**

1) Could you please elaborate on how realistic the problem setting is? Especially concerning the noisy intervention setting [6, 28], how compatible is the paper's problem setup with these real-world scenarios?

2) Could you please elaborate on the connection between your proposed method and existing literature on learning mixture of Gaussians?

**Limitations:**

Yes

---

> ### Author Rebuttal · Authors · 2024-08-07
>
> We thank the reviewer for their time and thoughtful comments. Below, we answer the weaknesses and questions raised by the reviewer.
>
> >Weakness
>
> **Weakness 1: Experiment on the real-world dataset:**
>
> We have run our algorithm on a real-world protein signaling network dataset. Our method performs equivalently on all the metrics compared to the Oracle algorithm, which has access to the disentangled data, unlike us (please refer to the **Evaluation on real-world dataset** paragraph in the Global comment for details). Also, In the **additional pdf document** attached to the global comment, we provide the ground truth graph and the estimated graph from both algorithms.
>
> \
> \
> **Weakness 2: Experiment with Linear SEM with non-gaussian noise:**
>
> The main motivation for our work comes from the problem of causal discovery. In general, one can only identify the causal graph up to their Markov Equivalence Class using only observational data. Thus, we need interventional data to fully identify the causal graph.  The intervention targets are sometimes noisy, so we can only obtain a mixture of interventional data.
>
> However, Shimizu et al. [1] showed that observational data is **sufficient for learning the underlying causal graph when the data-generating process is a linear SEM with additive non-Gaussian noise** with no latent confounders.  They also proposed an algorithm (LINGAM)  that uses Non-Linear ICA over observational data to identify the causal graph. Thus, Linear SEM with non-gaussian noise doesn't require interventional data to identify the underlying causal graph. Hence, studying the mixture of interventional distribution for this framework is an interesting problem outside the scope of current work.
>
> \
> \
> >Questions
>
> **Question 1**:
>
> The work from [6,28] has shown that CRISPR gene editing technology has an off-target effect. Also, the off-target effect can be random, i.e., every off-target effect could be different. For example, Aryal et al. [2] have shown that the same gene editing experiment on mice embryos exhibited different off-target cleavage for different mice. Thus, the observed data can be a mixture of multiple off-target interventions as modeled in our work. To perform any downstream task, one needs to identify the unknown intended target and disentangle the mixture distribution, which is also the main goal of our work.
>
> \
> \
> **Question 2**:
>
> In our work, we study the problem of disentangling a mixture of unknown interventions on Linear SEM with Gaussian noise, a special case of learning Gaussian mixtures. We remark that learning Gaussian mixtures is a well-studied problem with a rich literature (check Section 2), and our work builds upon these existing results. To invoke existing learning Gaussian mixture results, one needs to show the separation between distributions corresponding to the mixture components which is non-trivial. In our case, the mixture components correspond to interventional distributions, and one of our main contributions is to show the separation between them. We wish to emphasize that this separation also depends on the type of interventions one performs, and in our work, through careful analysis, we show the mean and covariance of interventional distributions are well separated for a more general class of soft interventions. As a consequence of our separation result, we show that the intervention targets and parameters of the international distribution can be recovered.
> \
> \
> [1] Shohei Shimizu, Patrik O. Hoyer, Aapo Hyvärinen, and Antti Kerminen. A linear non-gaussian acyclic model for causal discovery. Journal of Machine Learning Research, 7(72):2003–2030, 2006
>
> [2] N.K. Aryal, A.R. Wasylishen, and G. Lozano. Crispr/cas9 can mediate high-efficiency off- target mutations in mice in vivo. Cell Death and Disease, 9, 2018

---

> > ### Comment · Reviewer_nhPx · 2024-08-12
> > **Response to the Author**
> >
> > Thank you for your response. The authors have partially addressed my concerns, thus I will keep my rating.

---

### Official Review · Reviewer_mSnK · 2024-07-13

**Soundness:** 3
**Presentation:** 3
**Contribution:** 2
**Rating:** 5
**Confidence:** 3

**Summary:**

The paper proposes linear structural equation models with additive Gaussian noise to address the challenge of disentangling mixed interventional and observational data. The problem is highly relevant to real-world applications with mixed data.

**Strengths:**

* The theoretical framework is robust, with clear assumptions and derivations. The paper provides theoretical guarantees on the identifiability of mixture parameters.

* The key idea is clearly explained and easy to follow.

**Weaknesses:**

* The use of linear structural equation models is well-established in recent literature. Implementing SEMs with unknown interventions is not a novel approach.

* The identifiability guarantees rely on the assumption of soft interventions. This assumption may restrict the broader applicability of the model.

* The proposed method lacks sufficient experimental support. And it seems that the proposed method is primarily validated through simulations in the experiments, is it possible to provide real-world data examples?

**Questions:**

See Weaknesses.

**Limitations:**

* The method focuses on linear structural equation models with additive Gaussian noise only, and the theoretical guarantees rely on the linear- SEM assumptions, which may limit the method's applicability in practice.

---

> ### Author Rebuttal · Authors · 2024-08-07
>
> We thank the reviewer for their time and thoughtful comments. Below, we answer the weaknesses raised by the reviewer.
>
>
> **Weakness 1**:
>
> While we agree that the linear SEMs with unknown interventions are a relatively well-studied problem, we wish to highlight that our setting is different as we work with a mixture of multiple unknown interventions. Given such a mixture, our goal is to disentangle this mixture and recover the parameters corresponding to the mixture components. **To the best of our knowledge, no other work exists that aims to solve this problem of disentangling the mixture of unknown interventional distributions in the setting of linear SEMS with Gaussian noise**. Furthermore, in this setting, our work lays out the first theoretical foundation on the identifiability of the individual components in the mixture.
>
> We would like to restate the fundamental importance of this setup in the causal discovery literature (also stated in the Introduction, Lines 40-46 of the main paper). Shimizu et al. [1] showed that observational data is sufficient for learning the underlying causal graph when the data-generating process is a linear SEM with additive non-Gaussian noise with no latent confounders. However, in the same setting with Gaussian noise, the causal graph is only identifiable up to its Markov Equivalence Class (MEC). Thus, performing interventions (possibly noisy) is necessary to identify the causal graph, making it an interesting framework for our problem.
>
> We also wish to remark that some prior works do study a more general problem of disentangling the mixture of multiple unknown directed acyclic graphs. However, there are no theoretical guarantees about the identifiability of individual components in this setting (please refer to the Mixture of DAGs and Intervention paragraph in Section 2 of the main paper).
>
> \
> \
> **Weakness 2**:
>
> Soft interventions are a very general form of intervention that subsumes most of the widely studied interventions in the literature, such as shift, stochastic do, and do (please also see Lines 134 to 139 for this specialization). Therefore, we believe that studying soft interventions would indeed have a much broader applicability.
>
> \
> \
> **Weakness 3**:
>
>  We have run our algorithm on a real-world protein signaling network dataset. In summary, our method performs equivalently on all the metrics compared to the Oracle algorithm, which has access to the disentangled data (please refer to the **Evaluation on real-world dataset** paragraph in the Global comment for details). Also, in the **additional pdf document** attached to the global comment, we provide the ground truth graph and the estimated graph for both the algorithms. The causal graph estimated by our algorithm is very similar to the graph estimated by the oracle.
>
>
> Also, we have added results with a new version of our algorithm (Mixture-UTIGSP), where we automatically search for the number of components in the mixture (please see global comment **Automated Component Selection** for details). Evaluating the “half-setting,” where the number of components in the mixture is (num nodes +1)/2, we observe that:
> 1. The number of components found by this method is also close to the correct value.
> 2. The parameter estimation error goes down to zero for all the nodes, unlike Fig 1d in the main paper. The other metrics, like SHD, still have the same decreasing trend.
> 3. Please refer to Figure 2 in the **additional pdf document** attached to the global comment for the parameter estimation and SHD plot before (Fig1d and 1f from the main paper) and after this improvement.
>
>
> **Limitations 1**:
>
> We agree that the linear SEM with Gaussian noise is restrictive and may not apply to many real-world settings.
> However, even with these assumptions, the problem we study in our work is non-trivial and poses several challenges.
> Extending our results beyond the linear SEM setting would help broaden the applicability of our results. We believe that studying these questions is a very important future research direction, and we would like to consider our work as a first step in this direction. Also, in answer to the **Weakness 1** above, we motivated our choice to study Linear SEM with Gaussian noise.

---

> > ### Author Response · Authors · 2024-08-13
> >
> > As the deadline approaches, we would greatly appreciate hearing from you. Please let us know if you need any further clarifications.

---

> > ### Comment · Reviewer_mSnK · 2024-08-13
> >
> > I thank the authors for the detailed responses, which address some of my concerns, I will maintain my score.

---

### Official Review · Reviewer_583w · 2024-07-15

**Soundness:** 3
**Presentation:** 3
**Contribution:** 2
**Rating:** 6
**Confidence:** 4

**Summary:**

This paper considers a setup where an intervention results in obtaining iid data from a mixture of multiple interventional data and observational data and the goal is to learn the mixing weights and the resulting interventional distributions. Particular focus is on the linear SEM setting with Gaussian noise where interventions are allowed to be soft with some restrictions. The goal is to then learn the mixing weights and the interventional distribution (all multivariate Gaussians) parameters without knowledge of the causal graph. Given the large amount of literature on learning mixtures of Gaussians, it only remains to be shown that the individual components are well-separated in terms of changes in the interventional distributions.

**Strengths:**

Interventions in the real world turn out to be messy and this paper continues on the thread of research that deals with unknown interventions. The paper is well-written with a clear flow of ideas. The numerical evaluations are also thorough.

**Weaknesses:**

While I understand that this is the first step in learning mixtures of interventions, I believe that it has limited novelty both conceptually and technically since it largely follows from existing work on learning Gaussian mixtures. The considered setup also assumes unconfoundedness on top of linear SEMs and Gaussian noise. Few more detailed questions follow in the next section.

The experimental evaluation is also limited to simulation data which don't completely endorse the validation of the algorithm. See next section on suggestions/questions.

The writing can be more careful. There are typos in the main assumption (4.1) and there's some confusing notation about the high probability delta and the difference in variance delta_i.

**Questions:**

1. Theorem 4.1 is an existence theorem and in the experimental section, EM is used to learn the parameters. Given that the polynomial time guarantee includes information about A, how would we claim to not requiring knowledge of the causal graph?
2. While misspecifying k for the 'half' interventions case is indeed a feature, is there an improvement if k is correctly specified. Currently, it's not completely clear to me that the error goes down in (d).
3. Is there any intuition for why when SHD and Jaccard similarity metrics improve, the parameter estimation error still does not? See nodes = 8 in half interventions case.

**Limitations:**

The authors have addressed limitations in a separate section.

---

> ### Author Rebuttal · Authors · 2024-08-07
>
> We thank the reviewer for their time and thoughtful comments. Below, we answer the weaknesses and questions raised by the reviewer.
>
> > **Weakness**:
>
> **Limited Novelty**:  While we agree that our work builds on top of existing work but doesn’t follow immediately from them. To invoke these existing results on learning Gaussian mixtures, we do indeed make some non-trivial contributions, which we highlight below:
> 1. To invoke existing learning Gaussian mixture results, one needs to show the separation between distributions corresponding to the mixture components. In our case, the mixture components correspond to interventional distributions, and it is non-trivial to show separation between them. We wish to emphasize that this separation depends on the type of interventions one performs. Through careful analysis, our work shows that the mean and covariance of interventional distributions are well separated for a more general class of soft interventions.
> 2. In addition to the above, one of our other major contributions comes in modeling a real-world setting. Although the contributions in our work are a first step towards solving the practical problem, nevertheless, we believe it is an important first step.
>
> **Experimental Evaluation**: Please see the answer to Question 2. We have also added a new experiment in which we automatically select the number of components that improves the parameter estimation error in Fig 1d of the main paper (see answer to Question 3 below).
>
>
> **Writing Comment**: We thank the reviewer for pointing this out. Yes, the variance $\delta_i$ and the probability $\delta$ are different. We will update the main paper to incorporate the reviewer’s suggestion.
>
> \
> \
> > **Questions**:
>
> **Question 1**:
> Please note that Theorem 4.1 states that the **sample complexity** of the algorithm is polynomial in the norm of the adjacency matrix (A) but not the runtime as you stated. **The algorithm doesn’t explicitly use the knowledge of the adjacency matrix “A”**. The dependence of sample complexity on "A" characterizes the problem difficulty, i.e., depending on the problem, we will require a different number of samples to get the desired accuracy. Also, the algorithm is consistent, i.e., it will recover the correct parameters under an infinite sample limit.
> \
> \
> **Question 2**:
> 1. Fig 1a-c shows the setting when the number of components is correctly specified, i.e., k=num_node +1, and the number of components in the mixture is also num_nodes +1. In this case, we see a clear improvement in the parameter estimation error (Fig 1a). Fig 1d-f corresponds to the setting where the number of components is misspecified, i.e., k=num_node+1 and number of components = num_node/2+1. This is mainly due to mispecified number of components in the mixture. Below, we fix this.
> 2. **Automated Component Selection**: We have added results with a new version of our algorithm (Mixture-UTIGSP), where we automatically search for the number of components in the mixture (please see global comment **Automated Component Selection** for details). Evaluating the “half-setting,” where the number of components in the mixture is (num nodes +1)/2, we observe that:
>     1. The number of components found by this method is also close to the correct value.
>     2. The parameter estimation error goes down to zero for all the nodes, unlike Fig 1d in the main paper. The other metrics, like SHD, still have the same decreasing trend.
>    3. Please refer to Figure 2 in the **additional pdf document** attached to the global comment for the parameter estimation and SHD plot before (Fig1d and 1f from the main paper) and after this improvement.
>
>
> **Question 3**:
> The parameter estimation error is high in the second setting (num intervention = half, Fig 1d-f of main paper) due to the misspecified number of components (k=num nodes+1). To remove this problem, we have modified our algorithm to automatically select the correct number of components in the mixture (please see **Automatic component selection** paragraph in Global comments and answer to Question 2). In summary, with the modified algorithm, the parameter estimation error goes to zero, Jaccard similarity increases, and SHD decreases to zero as the sample size increases.

---

> > ### Comment · Reviewer_583w · 2024-08-13
> > **Response to rebuttal**
> >
> > Thanks for the response and apologies for my delayed response.
> > Regarding limited novelty, I am not convinced yet mainly because I don't see non-trivial cases where separation wouldn't hold. Perhaps it would be more constructive to consider a non-trivial example of intervention in your model where separation does not hold? I understand this request comes late.
> > Regarding Question 1 - Yes, I meant to write sample complexity. This is convincing since you're not really claiming polynomial sample complexity but just identifiability so the mismatch in the theorem statement and experimental algorithm is ok.
> >
> > Experimental Evaluation -
> > The authors have responded with a thorough evaluation including experiments on a real-world dataset. A question regarding the automatic component selection algorithm - how are the cutoff percentages decided?
> >
> > I still find the novelty issue a minus but the experimental evaluation is now thorough for me to increase my score.

---

> ### Author Response · Authors · 2024-08-13
>
> We thank the reviewer for their response and for acknowledging that the experimental evaluation is thorough now. Below, we answer the concerns and questions in detail:
>
> >**Limited Novelty**
>
> Here is an example where there is no separation between the parameters in the mixture.
> Let the system consist of three nodes $X, Y$ and $Z$, the corresponding SEM is defined as:
>
> $X= N(0,1)$
>
> $Y=X$ and
>
> $Z=X-Y+N(0,1)$
>
>
> Let the mixture distribution consist of two components: a) *observational* and b) *stochastic do* intervention on node $Z$. Here, we keep the noise distribution the same as before post-intervention, i.e., we set $Z = N(0,1)$. We can show that the mean and covariance of both components are the same. **Thus, there is zero parameter separation in spite of the adjacency matrix being different for both components**.
>
> We agree that the above constructed example violates the faithfulness assumption. Thus, we needed a careful analysis to capture such nuances and other complexities in the parameter separation calculation. Our lower bound of parameter separation in Lemma 5.1 captures this i.e., parameter separation is zero for the above example since $f(B, D)=\lambda_{min}^{2}(D)/4||I-A||_{F}^{4} = 0$  since the noise covariance matrix $D$ has one of the entries as zero and the other parameters $\delta_i=\delta_j=\gamma_i=\gamma_j=0$ (see Appendix A2 for exact expression of $f(B, D)$).
>
> Also, it is somewhat intuitive that for non-degenerate cases, the parameters will be separated, but the exact dependence on the parameters of intervention was not clear. In Lemma 5.1, we show that the lower bound of the parameter separation is a polynomial function of the parameters of intervention ($||c_i||$, $\delta_i$ and $\gamma_i$’s). This also enables us to show the **polynomial sample complexity of the existence algorithm** in Theorem 5.2.
>
>
>
>
>
> \
> \
> >**Threshold for Automatic Component Selection:**
>
> Currently, we manually set the percentage threshold to a fixed value (7% in our simulation experiments, selected arbitrarily). As we increase the number of components in the mixture model the log-likelihood of the model is expected to increase due to increasing model capacity. However, we expect that as we reach the correct number of components any further increase in the number of components will most likely lead to minimal change in the log-likelihood. Thus, we chose to keep the threshold a small number close to zero.

---

> > ### Comment · Reviewer_583w · 2024-08-13
> > **Final response**
> >
> > Thanks for the example and the clarification on the automatic component selection. Regarding the latter, I am not sure if this is then generalizable beyond the examples considered? I have increased my score accordingly.

---

> > > ### Author Response · Authors · 2024-08-13
> > >
> > > We thank the reviewer for the question regarding the generalizability of automatic component selection due to manual thresholding. Manual thresholding was mainly chosen due to the short time constraint for the rebuttal. However, in the final revision, we will try to include other well-known methods for model selection, like the BIC criterion, that would generalize to other datasets.

---

### Author Rebuttal · Authors · 2024-08-07

Below, we answer some of the common weaknesses or questions raised by multiple reviewers:



> Reviewer 583w

**Automatic component selection**:

In our Algorithm-1 (Mixture-UTIGSP), we allow for the misspecification of the correct number of components in a mixture. By default, the number of components is set as num node+1. However, this default setting could lead to errors in identifying the component's parameters, as shown in Fig 1d of the main paper. To address this issue, we have included results from a new version of our algorithm (Mixture-UTIGSP) that automatically searches for the number of components in the mixture. We do so by:
1. First, a different Gaussian mixture model should be fitted for all possible numbers of components, i.e., 1 to "num node+1".
2. num_fwd_component = Start from the model with 1 component, iterate to the model with the "num node +1" component, and stop where the change in the log-likelihood of the mixture model drops below a cutoff percentage.
3. num_bwd_component = Start from the “num node + 1” component model, iterate to the model with 1 component, and stop where the change in the likelihood of mixture increases above a cutoff percentage.
4. Number of component = (num_fwd_component+num_bwd_component)//2

Multiple other strategies, such as the BIC criterion or different variations of the above algorithm, could automatically select the number of components in the mixture, but we leave that exploration to future work.


We have rerun our experiment on the “half"-setting (where the true number of components in the mixture is (num nodes +1)/2) with this new automated selection method. Unlike Fig 1d in the main paper, where the parameter estimation doesn't improve much due to misspecified parameters, we observe that the parameter estimation errors go down to zero for all the nodes. Moreover, the other metrics, like Jaccard Similarity and SHD, still respectively show the same increasing and decreasing trend. Also, the estimated number of components becomes more accurate as the sample size increases. Please see Figure 2 in the **additional PDF document** attached for a comprehensive comparison of the parameter estimation and SHD plot before and after this improvement. For this experiment, we only utilized Step 3 of the modified algorithm mentioned earlier to determine the number of components.

\
\
> Reviewer 583w, mSnK, nhPx, SSWb

**Evaluation on real-world dataset**:

To demonstrate real-world applicability, we evaluate our method on the Protein Signaling dataset [1]. The dataset is collected from flow cytometry measurement of 11 phosphorylated proteins and phospholipids and is widely used in causal discovery literature [2,3]. The dataset consists of 5846 measurements with different experimental conditions and perturbations. Following Wang et al. [2], we define the subset of the dataset as observational, where only the receptor enzymes were perturbed in the experiment. Next, we select other 5 subsets of the dataset where a signaling protein is also perturbed in addition to the receptor enzyme. The observational dataset consists of 1755 samples, and the 5 interventional datasets have 911, 723, 810, 799, and 848 samples, respectively.  The table below summarizes the performance of our algorithm with the oracle (UTGSP), which already has access to disentangled data. For this experiment, our algorithm uses the automatic selection criteria for selecting the number of components in the mixture (*see Automatic component selection paragraph above*)

|  				| **Estimated vs Actual #component**  | **Jaccard Similarity** | **SHD** |
|--------------|----------------------------------|--------------------------------| -----------------|
|Ours (Mixture-UTIGSP)         |	7 (estimated) vs 6 (actual) | 0.08 | 17.6 +/- 1.0 |
|Oracle (UTIGSP) 	  	| NA | 0.09 | 17.4 +/- 1.0 |

The total number of nodes in the underlying causal graph is 11. Thus, the maximum possible component in the mixture is 12 (11 interventional and one observational). In the mixture dataset described above, we have 6 components (1 observational and 5 interventional). Our method automatically recovers 7 components from the mixture, close to the ground truth 6. Next, we give the disentangled dataset from the first step of our algorithm to identify the unknown target. Though the Jaccard similarity of the recovered target is not very high (0.08, where the maximum value is 1.0), it is similar to that of Oracle (UTGSP). This shows that it is difficult to identify the correct intervention targets even with correctly disentangled data. Also, the SHD between the recovered graph and the widely accepted ground truth graph for Mixture-UTIGSP (ours) and UTIGSP (oracle) is very close.

In the **additional pdf document** attached to the global comment, we provide the ground truth graph and the estimated graph from both algorithms. The causal graph estimated by our algorithm is very similar to the graph estimated by the oracle.

\
\
[1] K. Sachs, O. Perez, D. Pe’er, D. A. Lauffenburger and G. P. Nolan. Causal protein-signaling networks derived from multiparameter single-cell data. Science 308.5721 (2005): 523-529.

[2] Y. Wang, L. Solus, K. Yang, and C. Uhler. Permutation-based causal inference algorithms with interventions. In Advances in Neural Information Processing Systems, pages 5822–5831, 2017.

[3] Chandler Squires, Yuhao Wang, and Caroline Uhler. Permutation-based causal structure learning with unknown intervention targets. Proceedings of the Thirty-Sixth Conference on Uncertainty in Artificial Intelligence, UAI 2020

---

### Decision · Program_Chairs · 2024-09-25

**Decision:**

Accept (poster)

**Comment:**

The reviewers are recommending acceptance after the rebuttal, although some had reservations about the contribution being on the minor end since the method heavily draws from disentangling a mixture of Gaussians. The additive noise assumption is also seen as a weakness of the analysis. However, I believe the sample-complexity results are much needed in the causal discovery literature and might justify the needed ANM assumption. Reviewer SSWb gave an 8 with a confidence of 5, but I do not think their review content justifies this high score/confidence pair. Looking at the other scores, this paper is a bit above borderline in its current form in my opinion.